# LEARNING RESIDUE LEVEL PROTEIN DYNAMICS WITH MULTISCALE GAUSSIANS

**Mihir Bafna**[1,3]    **Bowen Jing**[1]    **Bonnie Berger**[1,2]

[1] CSAIL, Massachusetts Institute of Technology
[2] Dept. of Mathematics, Massachusetts Institute of Technology
[3] Liquid AI

{mihirb14, bjing, bab}@mit.edu

## ABSTRACT

Many methods have been developed to predict static protein structures, but understanding structure *dynamics* is what is essential for elucidating biological function. While molecular dynamics (MD) simulations remain the *in silico* gold standard, its high computational cost limits scalability. We present DYNAPROT, a lightweight, SE(3)-invariant framework that predicts rich descriptors of protein dynamics directly from static structures. By casting the problem through the lens of multivariate Gaussians, DYNAPROT estimates dynamics at two complementary scales: (1) per-residue marginal anisotropy as $3 \times 3$ covariance matrices capturing local flexibility, and (2) joint scalar covariances encoding pairwise dynamic coupling across residues. From these dynamics outputs, DYNAPROT achieves high accuracy in predicting residue-level flexibility (RMSF) and, remarkably, enables reasonable reconstruction of the full covariance matrix for fast ensemble generation. Notably, it does so using orders of magnitude fewer parameters than prior methods. Our results highlight the potential of direct protein dynamics prediction as a scalable alternative to existing methods.

## 1 INTRODUCTION

Proteins rarely exist in static conformations. Due to interactions with ligands, other biomolecules, and external factors such as temperature and pH, protein structures continuously fluctuate. Many enzymes rely on loop motions or domain rearrangements to form catalytically active sites (Zinovjev et al., 2024), allostery often involves shifting backbone or side chain conformations propagating signals over long distances (Yu and Koshland Jr, 2001), and even membrane proteins such as GPCRs switch between inactive and active conformational states essential for signal transduction (Zhang et al., 2024). Clearly, protein structures are *dynamic*. Understanding these dynamics is central for mechanistic insight and, potentially, the design of functions (Guo et al., 2025; McCammon, 1984).

Capturing this ensemble-level behavior computationally has long been the domain of molecular dynamics (MD). MD simulates the time evolution of atoms under a force field, generating high-resolution conformational trajectories from which fluctuations, covariances, and time-dependent observables can be derived. MD remains the *in silico* gold standard for protein dynamics, offering fine-grained, physically grounded insights (Shaw et al., 2010; Hollingsworth and Dror, 2018; Childers and Daggett, 2017). However, it comes with an enormous computational cost: simulating 100 ns of dynamics for a single protein can take days or weeks on specialized hardware. This limits its scalability, especially for proteome-wide applications or tasks requiring real-time dynamics estimates.

Recent work has explored the use of deep learning to approximate and accelerate this process. Generative modeling-based methods like ALPHAFLOW (Jing et al., 2024) repurpose AlphaFold2 (Jumper et al., 2021) under a flow matching paradigm to sample protein conformations. Along this vein, Lewis et al. (2024) recently introduced BIOEMU as a large-scale diffusion model pretrained on PDB (Burley et al., 2017) and AFDB structures, and fine-tuned on 200 ms of MD data, to efficiently generate protein conformations. Other methods, like MSA subsampling, make inference-time adjustments to the MSA input of AlphaFold2, yielding the structural ensembles (Del Alamo

et al., 2022; Wayment-Steele et al., 2024; Stein and Mchaourab, 2022). Still, all of these approaches necessitate large scale PDB pretraining and suffer from inference-time computational overhead, requiring multiple stochastic forward passes to generate meaningful structural diversity. Moreover, while these ensembles can be used to approximate protein dynamics, generating them remains time intensive, and the full ensemble of diverse conformations may not always be necessary. In many practical settings, compact and interpretable representations of dynamics often suffice. This motivates the need for models that can *explicitly* predict such dynamics descriptors without relying on *implicit* dynamics learners like expensive ensemble generation methods.

Current explicit dynamics predictors, like FLEXPERT3D (Kouba et al., 2024), resort to predicting simple collective variables like per-residue RMSF, a scalar quantifying each residue's positional fluctuation. RMSF is widely used due to its simplicity and interpretability, but it is fundamentally limited: it captures only the magnitude of local motion and discards directionality and residue-residue coupling. Similarly, Wayment-Steele et al. (2025) trained DYNA-1 to predict labels of $\mu$s–ms motion by cleverly exploiting missing chemical shift assignments as hidden observables in NMR ensembles, but these predictions also remain scalar and lack directionality. A different example of an explicit dynamics predictor is Normal Mode Analysis (NMA), a classic technique that approximates dynamics by identifying low frequency eigenmodes to describe the largest movements (Cui and Bahar, 2005; Skjaerven et al., 2009). NMA does not learn from data however, and instead operates solely on the input PDB structure. It can estimate the principal global directions of motion and offers insights into collective flexibility, but is sensitive to input structure quality and fails to adequately capture local anisotropy or conformational heterogeneity (Ma, 2005). This raises a natural question:

> Can we design models that lie on the Pareto frontier of expressiveness and efficiency—capturing rich dynamic behavior without incurring the cost of sampling or simulation?

We introduce DYNAPROT, a lightweight, interpretable, and expressive framework for predicting protein dynamics through the lens of Gaussian distributions over structure (Section 2). Specifically, DYNAPROT predicts: (1) per-residue marginal anisotropy as $3 \times 3$ covariance matrices capturing local dynamics while encompassing RMSF, and (2) joint scalar $N \times N$ covariances encoding pairwise dynamic coupling across residues. Remarkably, while DYNAPROT was not explicitly trained to directly model the full $3N \times 3N$ joint distribution, we find that its marginal and pairwise outputs can be composed into a reasonable approximation (Section 3.4), enabling extremely fast ensemble generation in $\mathbb{R}^{3N}$. DYNAPROT is trained on only $\sim$1,000 MD-derived proteins,

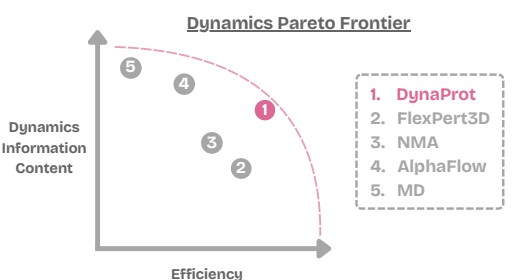

Figure 1: Dynamics methods information content vs. efficiency.

without large-scale pretraining on PDB structures, and improves upon Normal Mode Analysis (NMA) in both predictive accuracy and efficiency, while remaining dramatically smaller and faster than existing ensemble generation approaches.

To our knowledge, DYNAPROT is the first model to *explicitly* learn both marginal and pairwise Gaussian representations of protein dynamics, and the first to predict the full $3N \times 3N$ covariance structure—akin to NMA—in a data-driven, learnable fashion, rather than relying solely on analytical approximations or less informative per-residue fluctuations.

## 2 GAUSSIAN REPRESENTATION OF DYNAMICS

We propose a perspective for modeling protein dynamics through distributions over atomic coordinates, relying on tractable approximations such as Gaussians. Formally, we model a protein structure with $N$ residues as a random variable $\boldsymbol{X} \in \mathbb{R}^{3N}$, where each residue contributes the three-dimensional Cartesian coordinates of its $C_\alpha$ atom. While this coarse-grained representation omits side-chain flexibility, it enables scalable modeling of backbone dynamics, which is the scope of our work. We consider an *ensemble* to be $T$ independent samples after RMSD alignment. The

distribution over conformational states is then represented as a multivariate normal distribution:

$$\boldsymbol{X} \sim \mathcal{N}(\boldsymbol{\mu}, \boldsymbol{\Sigma}_{\text{joint}}), \quad \boldsymbol{\mu} \in \mathbb{R}^{3N}, \; \boldsymbol{\Sigma}_{\text{joint}} \in \mathbb{R}^{3N \times 3N} \tag{1}$$

Here, $\boldsymbol{\mu}$ corresponds to the average (or equilibrium) structure—typically the minimum energy conformation—and $\boldsymbol{\Sigma}_{\text{joint}}$ captures the full covariance across all $C_\alpha$ positions, encoding both local fluctuations and long-range correlated motions. This joint covariance matrix theoretically encodes all second-order information about the protein's dynamics: from it, one can derive a wide range of collective variables including principal components (PCs) of motion, residue-residue distance variances, and global flexibility metrics. The Gaussian formulation provides a principled way to decompose protein dynamics across different levels of granularity (Figure 2 and Table 1), depending on the modeling objective. At the local level, the marginal distribution

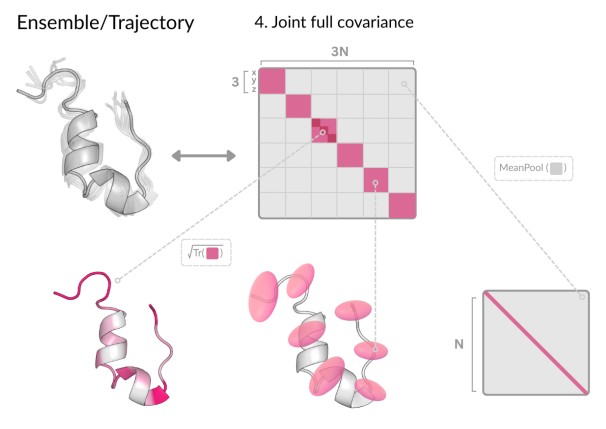

Figure 2: Gaussian view of dynamics.

(Figure 2.2) for a single residue $i$ is obtained by integrating out all other residue coordinates: $p(\boldsymbol{x}_i) = \int p(\boldsymbol{x}_1, \ldots, \boldsymbol{x}_n) \, d\boldsymbol{x}_{\neg i}$, where $d\boldsymbol{x}_{\neg i} := \prod_{j \neq i} d\boldsymbol{x}_j$. This results in a 3D Gaussian distribution over the $C_\alpha$ coordinates of residue $i$:

$$\boldsymbol{X}_i \sim \mathcal{N}(\boldsymbol{\mu}_i, \boldsymbol{\Sigma}_{\text{marginal}}^{(i)}), \quad \boldsymbol{\mu}_i \in \mathbb{R}^3, \; \boldsymbol{\Sigma}_{\text{marginal}}^{(i)} \in \mathbb{R}^{3 \times 3} \tag{2}$$

where $\boldsymbol{\Sigma}_{\text{marginal}}^{(i)}$ is the $3 \times 3$ diagonal block of $\boldsymbol{\Sigma}_{\text{joint}}$. These marginals can be interpreted as *Gaussian blobs* encoding anisotropic local fluctuations—i.e., spatial variance of where each residue may reside.

Notably, this formulation allows for simple derivation of scalar flexibility metrics such as the *root-mean-square fluctuation (RMSF)* as $\text{RMSF}_i = \sqrt{\text{Tr}(\boldsymbol{\Sigma}_{\text{marginal}}^{(i)})}$. RMSF (Figure 2.1) represents a simple notion of dynamics: a single scalar per residue quantifying positional fluctuation. However, it discards directional and covariance structure captured by the full marginal.

To capture dynamics beyond residue-local fluctuations, we also consider a covariance matrix $\boldsymbol{C} \in \mathbb{R}^{N \times N}$ of scalar pairwise coupling (Figure 2.3). Each entry $\boldsymbol{C}_{ij}$ summarizes the dynamical coupling between residues $i$ and $j$, typically computed as a scalar projection of the corresponding $3 \times 3$ block in the full joint covariance: $\boldsymbol{\Sigma}_{\text{joint}}[3i : 3i + 3, 3j : 3j + 3]$. We choose MeanPooling as the scalar projection to compute each $\boldsymbol{C}_{ij}$. This compact representation enables efficient modeling of residue-residue coupling.

Table 1: Taxonomy of protein dynamics representations under a Gaussian view.

| Level | Description | Notation | Space | Captures |
|-------|-------------|----------|-------|----------|
| 1 | Per-residue scalar (i.e. RMSF) | $\sqrt{\text{Tr}(\boldsymbol{\Sigma}_{\text{marginal}}^{(i)})}$ | $\mathbb{R}^N$ | Magnitude of fluctuation per residue |
| 2 | Per-residue full (Gaussian blob) | $\boldsymbol{\Sigma}_{\text{marginal}}^{(i)}$ | $\mathbb{R}^{N \times 3 \times 3}$ | Anisotropic local covariance per residue |
| 3 | Joint scalar (pairwise coupling) | $\boldsymbol{C}_{ij}$ | $\mathbb{R}^{N \times N}$ | Scalar covariance across all residues |
| 4 | Joint full covariance | $\boldsymbol{\Sigma}_{\text{joint}}$ | $\mathbb{R}^{3N \times 3N}$ | Full spatial covariance across all residues |

DYNAPROT focuses on levels 2 and 3 of this hierarchy—explicitly predicting both $3 \times 3$ marginal Gaussians per-residue and a $N \times N$ matrix of residue-residue couplings. As noted before, from the $3 \times 3$ marginals, we can easily derive RMSF (level 1). Interestingly, utilizing both the marginals and the pairwise coupling, we can retrieve a reasonable approximation of the full joint $3N \times 3N$ (level 4; Section 3.4). This design strikes a balance between local interpretability and global coordination, while avoiding the intractability of directly learning the full joint covariance.

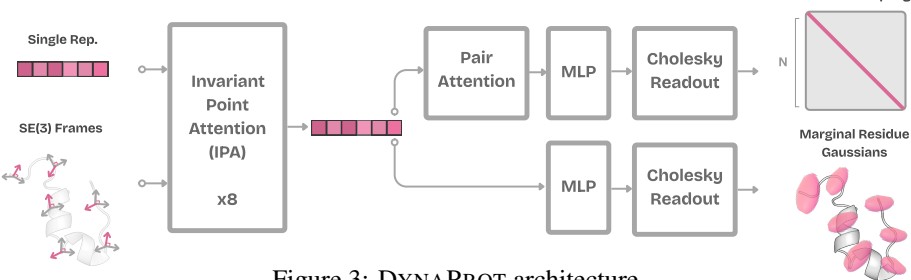

Figure 3: DYNAPROT architecture.

## 3 METHOD

### 3.1 DYNAPROT OVERVIEW

DYNAPROT (Figure 3) consists of two models, each taking as input a protein structure but designed to *explicitly* capture different granularities of protein dynamics: (i) marginal Gaussian blobs per residue (Section 3.2), and (ii) pairwise covariance across residues (Section 3.3). DYNAPROT is given the input structure as a set of local $C_\alpha$ residue frames. The frames are denoted $\{T_i\}_{i=1}^N$, where each frame $T_i \in \mathrm{SE}(3)$ is parameterized by a rotation matrix $R_i \in \mathrm{SO}(3)$ and a translation vector $t_i \in \mathbb{R}^3$. Simply put, $T_i = (R_i, t_i)$ captures the local orientation and position of residue $i$. Additionally, an initial embedding layer is included to encode the amino acid sequence $s \in \mathbb{R}^{N \times D}$.

Both models share a common architectural backbone composed of eight Invariant Point Attention (IPA) blocks from the structure module of AlphaFold2 (Jumper et al., 2021). These blocks are designed to encode geometric relationships between residues while maintaining invariance to SE(3) transformations (global transformations do not affect the learned residue-level representations). The IPA backbone processes the set of residue frames and the sequence representation, outputting a learned representation for each residue $h \in \mathbb{R}^{N \times D}$. No pair representation is given as input to the model. The two models differ only in their readout layers, which we define in the proceeding sections.

### 3.2 LEARNING MARGINAL GAUSSIANS

After the input sequence representations and residue frames are processed through the IPA backbone, a simple MLP readout is used for marginal prediction. Given the hidden representation $h_i$ for each residue, the marginal readout outputs $\Sigma^{(i)}_{\mathrm{marginal}} \in \mathbb{R}^{3 \times 3}$, modeling the local position (xyz) covariance of residue $i$. These outputs are trained to match empirical marginal distributions derived from the MD data. Note that the mean of each Gaussian is not learned. Instead, we take the input structure's $C_\alpha$ coordinate $t_i \in \mathbb{R}^3$ as the fixed mean $\mu_i$ of the distribution: $\mu_i := t_i$, $X_i \sim \mathcal{N}(\mu_i, \Sigma^{(i)}_{\mathrm{marginal}})$. This assumption is motivated by the fact that the input structure usually corresponds to the experimentally determined (or AlphaFold-predicted) minimum energy conformation, and thus serves as a natural estimator of the ensemble mean. Consequently, the marginal prediction task reduces to learning the covariance matrices $\Sigma^{(i)}_{\mathrm{marginal}}$ alone.

**Marginal dynamics module.** Recall that covariance matrices are required to be symmetric and positive definite (SPD). Predicting all 9 elements of a $3 \times 3$ matrix would be overparameterized and does not guarantee SPD structure. Naively, one might consider symmetrizing an arbitrary matrix after predicting the 6 independent elements, but this only guarantees symmetry. Instead, we leverage the fact that any SPD matrix can be uniquely defined by its Cholesky factorization. Thus, we enforce SPD constraints directly by parameterizing the covariance via its Cholesky factor. Specifically, the model predicts the entries $\{a_j\}_{j=1}^6$ of a lower triangular matrix $L_i \in \mathbb{R}^{3 \times 3}$, enforces positivity along the diagonal with the Softplus activation function (Glorot et al., 2011), and recovers the covariance:

$$\Sigma^{(i)}_{\mathrm{marginal}} = L_i L_i^\top, \quad \text{where } L_i = \begin{bmatrix} \mathrm{softplus}(a_1) & 0 & 0 \\ a_2 & \mathrm{softplus}(a_3) & 0 \\ a_4 & a_5 & \mathrm{softplus}(a_6) \end{bmatrix} \quad (3)$$

This factorization ensures that the predicted covariance matrix is SPD by construction. Since SPD matrices lie on a Riemannian manifold with non-Euclidean geometry, using loss functions that respect

this structure is critical for meaningful comparison. Standard Euclidean distances (e.g., MSE or Frobenius norm) ignore the curvature of this space and can lead to unstable or distorted gradients (see ablations in Appendix A.3.2). We instead employ the log-Euclidean (or log-Frobenius) distance (Vemulapalli and Jacobs, 2015; Huang et al., 2015) that reflects the intrinsic geometry of the SPD manifold. The Bures-Wasserstein (Bhatia et al., 2019) distance can also be used, but we find the log-Frobenius distance to be more stable.

$$\mathcal{L}_{\text{LogFrob}} = \| \log(\boldsymbol{\Sigma}_{\text{pred}}) - \log(\boldsymbol{\Sigma}_{\text{true}}) \|_F^2, \quad \text{where } \log(\boldsymbol{\Sigma}) = \boldsymbol{Q} \log(\boldsymbol{\Lambda}) \boldsymbol{Q}^T \tag{4}$$

As Riemannian manifolds are "locally Euclidean", this loss applies the matrix logarithm mapping the SPD matrix to its tangent space where a Euclidean metric (canonical Frobenius norm) can be utilized.

## 3.3 LEARNING PAIRWISE DYNAMICS

Using the output representations $\boldsymbol{h}$ from the IPA backbone, the pairwise dynamics module produces a scalar-valued $N \times N$ covariance matrix $\boldsymbol{C}$, where each entry $\boldsymbol{C}_{ij}$ captures the dynamical coupling between residue pairs. These scalar couplings are derived from the full joint covariance matrix via averaging per block and trained to reproduce MD-derived pairwise fluctuations.

**Pairwise Dynamics Module.** To predict the global pairwise covariance structure, we first construct pairwise features for all residue pairs. For each pair $(i, j)$, we concatenate their residue-level embeddings $[\boldsymbol{h}_i \,\|\, \boldsymbol{h}_j] \in \mathbb{R}^{2d}$ and project them into a lower-dimensional space: $f_{ij}^{(0)} = W_{\text{proj}}[\boldsymbol{h}_i \,\|\, \boldsymbol{h}_j] \in \mathbb{R}^{d'}$.

We pass these features through a stack of AlphaFold-style pairwise attention blocks based on the Evoformer architecture (Jumper et al., 2021), which include triangle updates and residue-wise message passing. These operations are designed to model transitive and higher-order geometric dependencies across residue pairs, and have been shown to be highly effective in structure-aware tasks: $f_{ij}^{\text{attn}} = \text{PairwiseAttentionBlock}(f_{ij}^{(0)}) \in \mathbb{R}^{d'}$. The output $f_{ij}^{\text{attn}}$ serves as a "learned basis"" over the space of residue-residue covariance structure. These basis features are then mapped to scalars through an MLP head, yielding a covariance for each pair of residues: $z_{ij} = \text{MLP}(f_{ij}^{\text{attn}})$, $z_{ij} \in \mathbb{R}$, for $i \geq j$. Following the same procedure as Section 3.2, we enforce SPD constraints on this covariance matrix by populating the lower-triangle entries of $\boldsymbol{L} \in \mathbb{R}^{N \times N}$ with the values of $z_{ij}$ and applying the Softplus activation when $i = j$. Finally, the pairwise covariance matrix is reconstructed via Cholesky composition $\boldsymbol{C} = \boldsymbol{L}\boldsymbol{L}^T$ and again equation 4 is used for optimization.

## 3.4 LEARNING THE FULL JOINT FOR ENSEMBLE SAMPLING

**Joint reconstruction heuristic.** Given a predicted scalar coupling matrix $\boldsymbol{C} \in \mathbb{R}^{N \times N}$ and a set of per-residue marginal covariances $\{\boldsymbol{\Sigma}_{\text{marginal}}^{(i)} \in \mathbb{R}^{3 \times 3}\}_{i=1}^N$, we propose a heuristic to reconstruct an approximate full joint covariance matrix $\boldsymbol{\Sigma}_{\text{joint}} \in \mathbb{R}^{3N \times 3N}$.

Each marginal covariance $\boldsymbol{\Sigma}_{\text{marginal}}^{(i)}$ is SPD by construction, and thus admits a Cholesky factorization $\boldsymbol{\Sigma}_{\text{marginal}}^{(i)} = \boldsymbol{L}_i \boldsymbol{L}_i^\top$, where $\boldsymbol{L}_i \in \mathbb{R}^{3 \times 3}$. We then define a block-diagonal matrix $\boldsymbol{L}_{\text{marginal}} \in \mathbb{R}^{3N \times 3N}$ as $\boldsymbol{L}_{\text{marginal}} = \bigoplus_{i=1}^N \boldsymbol{L}_i$. By construction, $\boldsymbol{L}_{\text{marginal}}$ is lower triangular with positive diagonal entries, since each $\boldsymbol{L}_i$ satisfies these properties.

Drawing from the univariate identity $\text{Cov}(i, j) = \text{Corr}(i, j) \cdot \sigma_i \sigma_j$, we define the multivariate cross-covariance block between residues $i$ and $j$ as $\boldsymbol{\Sigma}_{\text{joint}}^{(i,j)} = \boldsymbol{L}_i \widetilde{\boldsymbol{C}}_{ij} \boldsymbol{L}_j^\top$. Here, the Cholesky factor $\boldsymbol{L}_i$ serves as a matrix square root of the covariance $\boldsymbol{\Sigma}_{\text{marginal}}^{(i)}$, analogous to standard deviation in the univariate case. And, $\widetilde{\boldsymbol{C}}$ is a correlation matrix found by standardizing $\boldsymbol{C}$. Using the Kronecker product, we can denote this heuristic cleanly as follows,

$$\boldsymbol{\Sigma}_{\text{joint}} = \boldsymbol{L}_{\text{marginal}} \left( \widetilde{\boldsymbol{C}} \otimes \boldsymbol{I}_3 \right) \boldsymbol{L}_{\text{marginal}}^\top \tag{5}$$

**Proposition 3.1** (SPD Closure). *Given marginal covariances $\{\boldsymbol{\Sigma}_{marginal}^{(i)} \in \mathbb{R}^{3 \times 3}\}_{i=1}^N$ and correlation matrix $\widetilde{\boldsymbol{C}} \in \mathbb{R}^{N \times N}$ to be symmetric and positive definite, then the reconstructed joint covariance $\boldsymbol{\Sigma}_{joint} = \boldsymbol{L}_{marginal}(\widetilde{\boldsymbol{C}} \otimes \boldsymbol{I}_3)\boldsymbol{L}_{marginal}^\top$ is also symmetric and positive definite.*

We refer the reader to Appendix A.1 for the proof. This approximation combines local anisotropic uncertainty with global correlation structure. While not exact, we find it reconstructs the joint covariance to a reasonable degree and serves as a useful tool for downstream ensemble generation.

**Ensemble sampling.** Given the reconstructed joint covariance $\boldsymbol{\Sigma}_{\text{joint}}$ and our assumption that the mean $\boldsymbol{\mu}$ corresponds to the coordinates of the input structure (e.g., the PDB), we have now retrieved our Gaussian distribution over conformations $\mathcal{N}(\boldsymbol{\mu}, \boldsymbol{\Sigma}_{\text{joint}})$. To sample from this distribution, we apply a multivariate generalization of the reparameterization trick used in univariate Gaussian sampling.

---

**Property 3.1** (Multivariate Gaussian Sampling). *Given $\mathcal{N}(\boldsymbol{\mu}, \boldsymbol{\Sigma})$, where $\boldsymbol{\Sigma} \in \mathbb{R}^{d \times d}$ is SPD and $\boldsymbol{\Sigma} = \boldsymbol{L}\boldsymbol{L}^{\top}$ is its Cholesky decomposition. Then,*

$$\boldsymbol{x} = \boldsymbol{\mu} + \boldsymbol{L}\boldsymbol{\epsilon}, \quad \boldsymbol{\epsilon} \sim \mathcal{N}(\boldsymbol{0}, \boldsymbol{I}_d) \quad \Rightarrow \quad \boldsymbol{x} \sim \mathcal{N}(\boldsymbol{\mu}, \boldsymbol{\Sigma})$$

---

Note that this sampling relies directly on the Cholesky factor (similar to a matrix square root), mirroring the scalar case (Appendix A.1). Utilizing DYNAPROT predictions and this heuristic, ensemble sampling becomes extremely fast with minimal computational overhead.

## 4 EXPERIMENTS

**Preprocessing.** We construct ground-truth dynamics labels from the ATLAS molecular dynamics dataset, which comprises 1,390 proteins selected based on structural diversity using the ECOD domain classification (Vander Meersche et al., 2024). Following ALPHAFLOW (Jing et al., 2024) for preprocessing consistency, we concatenate each of the three replicate simulations of 100 ns per protein and extract the $C_\alpha$ coordinates. From each ensemble, we compute the empirical full joint covariance matrix over time and extract the relevant dynamics labels ($3 \times 3$ marginals per residue and $N \times N$ residue coupling) as described in Section 2. We evaluate under two train/val/test split regimes. The primary matches ALPHAFLOW's (1265/39/82), while comparisons to FLEXPERT3D use DYNAPROT trained on their topology-based split (1112/139/139). For naming, we refer to DYNAPROT-M for the model trained for marginals, DYNAPROT-J for the coupling predictions, and DYNAPROT for both.

**Baselines.** For a faithful comparison, we mainly choose baseline methods that take a protein structure as input and predict dynamics descriptors either *implicitly* (AFMD+TEMPLATES) or *explicitly* (FLEXPERT3D, NMA). For NMA, we utilize the ProDy package (Zhang et al., 2021), specifically the Anisotropic Network Model instantiation. For a broader set of baselines, we also compare against some sequence based methods in Appendix A.3.3.

### 4.1 PREDICTING RESIDUE FLEXIBILITY

Since DYNAPROT-M is trained to predict marginal Gaussians per residue, it inherently captures residue-level flexibility, as RMSF is defined as the square root of the trace of each marginal covariance (see Section 2). To evaluate DYNAPROT-M's ability to recover this, we compare against what is, to our knowledge, the only method that *explicitly* predicts residue flexibility: FLEXPERT-3D. For fair comparison, we train and evaluate DYNAPROT-M under the same ATLAS train/val/test split defined in Kouba et al. (2024). Despite solving the more challenging task of predicting marginal anisotropy rather than scalar fluctuations alone, DYNAPROT-M achieves a sub-

Table 2: RMSF Pearson correlation ($r$) against ATLAS MD-derived RMSF (FlexPert test split). Median and 75th percentile reported.

| Method | RMSF $r$ ($\uparrow$) | # Params |
|---|---|---|
| DYNAPROT-M | **0.865 / 0.930** | **955 K** |
| FLEXPERT-3D | 0.830 / 0.899 | 1.2 B |
| NMA (ANM) | 0.697 / 0.784 | – |

stantially higher Pearson correlation with MD-derived RMSF (median $r = 0.865$, 75th percentile $r = 0.930$) than FLEXPERT-3D (Table 2), while using three orders of magnitude fewer parameters (955K vs. 1.2B) and without NMA as input. This allows DYNAPROT-M to generalize better while being more parameter efficient. See Appendix A.4 for DYNAPROT-M additional RMSF plots.

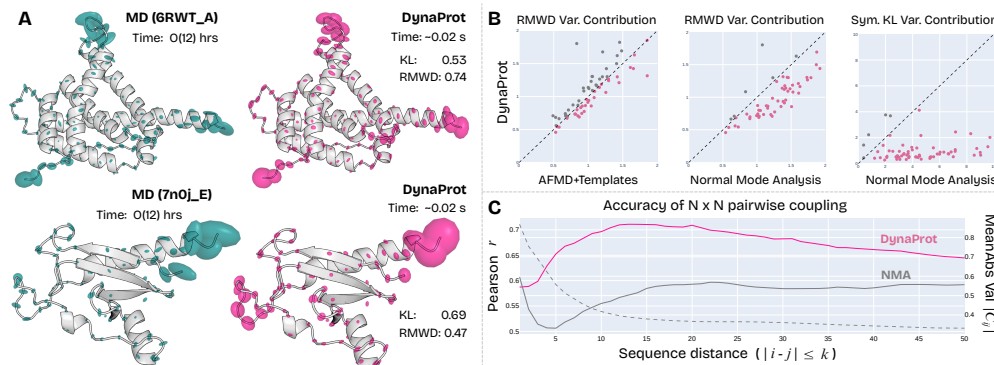

Figure 4: **DYNAPROT marginal Gaussian and residue coupling analysis**. **A.** Renderings of predicted marginal Gaussians compared to ATLAS MD constructed Gaussians (mean symmetric KL divergence and RMWD are reported). **B.** Joint distribution (within 75th percentile) of DYNAPROT performance vs. (AFMD+T, NMA). **C.** Band-wise Pearson correlation between predicted and ground-truth residue–residue coupling matrices as a function of sequence distance.

## 4.2 PREDICTING RESIDUE FULL ANISOTROPY

To assess the faithfulness of DYNAPROT-M's predicted marginals, we compare against both physics-based and learned ensemble methods. In practice, only NMA (ANM) is a feasible baseline, as ALPHAFLOW is prohibitively slow: A single 271-length protein (7lao_A) requires ∼7000 s, compared to ∼0.02 s for DYNAPROT-M (Table 3). Note that DYNAPROT-M predicts this directly, but for ALPHAFLOW, we first sample 250 structures per protein and then calculate the empirical covariance to define the marginal Gaussians. With NMA, we retrieve the full joint via the normal modes and extract the marginal block diagonals.

To quantify the accuracy, we compute the variance contribution of the *symmetric KL divergence* (see Appendix A.2) and the *root mean 2-Wasserstein distance* (RMWD) as described in Jing et al. (2024), compared to the ground truth marginal Gaussians computed from the ATLAS test set (AFMD split).

Despite being orders of magnitude faster and smaller (955k vs. 95M parameters), DYNAPROT-M achieves competitive accuracy. DYNAPROT-M attains a median RMWD of 1.18 and symmetric KL divergence of 0.91, both substantially better than NMA (1.45 and 4.56, respectively), and comparable to AFMD+TEMPLATES's 1.10 and 0.60.

Table 3: Comparison of methods on anisotropic blob prediction (ATLAS test split). Runtime for a length 271 protein (7lao_A). 25th %ile / Median reported (↓ is better).

| Method | RMWD Var | Sym. KL Var | # Params | Time |
|---|---|---|---|---|
| DYNAPROT-M | **0.84** / 1.18 | 0.53 / 0.91 | **955 K** | ∼**0.02** s |
| AFMD+T | 0.87 / **1.10** | **0.37** / **0.60** | 95 M | ∼7000 s |
| NMA (ANM) | 1.14 / 1.45 | 3.03 / 4.56 | – | ∼5.37 s |

Moreover, rather than relying solely on summary statistics, we also visualize the distributions (75th percentile) of RMWD and mean symmetric KL variance contributions across test set proteins (Figure 4B). These plots compare DYNAPROT-M to both AFMD+T and NMA on a per-protein basis. Points below the diagonal (highlighted in pink) indicate that DYNAPROT-M outperformed the method in question on that particular protein. From this, we see that DYNAPROT-M achieves comparable performance to AFMD+T. Notably, within the 75th percentile it often outperforms AFMD+T (examples visualized in Figure 4A,B) on RMWD variance contribution. Moreover, DYNAPROT-M significantly outperforms NMA across both RMWD and symmetric KL. This further corroborates DYNAPROT-M's ability to capture local anisotropic structure well despite being much smaller and faster than other methods.

### 4.2.1 ZERO-SHOT CRYPTIC POCKET DISCOVERY OF ADENYLOSUCCINATE SYNTHETASE

Beyond accuracy, DYNAPROT-M's marginals can also provide functional insight. Many proteins are considered to be undruggable as their *apo* form may not display a clear binding pocket. However, the druggable pocket may only become apparent after the drug is bound (*holo* form)– a so called "cryptic pocket." Identification of cryptic pockets is therefore an important task in drug discovery (Mou et al., 2025; Hollingsworth et al., 2019; Comitani and Gervasio, 2018).

As a case study, we sought to investigate DYNAPROT's ability in cryptic pocket identification for the enzyme adenylosuccinate synthetase, as it is known to exhibit a cryptic pocket (Meller et al.) and both the *apo* and *holo* forms are available in the PDB (`1ADE` / `1CIB`).

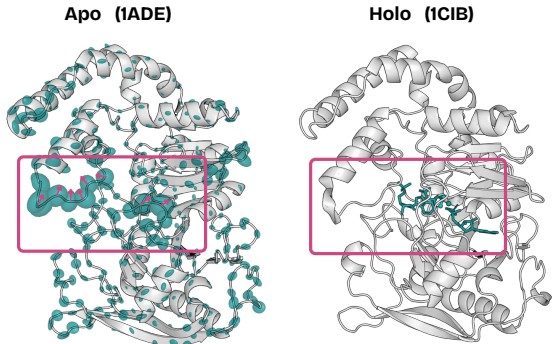

Figure 5: DYNAPROT-M predicted residue Gaussians (ellipsoids) overlaid the *apo* form.

We applied DYNAPROT-M to zero shot predict the marginal Gaussians on the *apo* form. When we look at the predictions, we notice two features (Figure 5). First, the residues with the largest variance are exactly those that encompass the binding pocket. Second, more than just the magnitude of the Gaussians, the directionality is consistent with a potential pocket opening motion (when compared with the *holo* form `1CIB`). These early results suggest the potential of DYNAPROT's utility in cryptic pocket discovery, but a systematic exploration is left for future work.

### 4.3 PREDICTING SCALAR COUPLING

To evaluate DYNAPROT-J in modeling residue–residue couplings, we compare its predicted $N \times N$ scalar covariance matrices against those derived from NMA, a classical method for capturing global structural dynamics. While DYNAPROT-J directly predicts these scalar coupling matrices, we construct comparable matrices for NMA by computing full $3N \times 3N$ anisotropic network models (using ProDy) from each test protein, and then projecting them into $N \times N$ scalar covariances as described in Section 2. We obtain per-residue correlation matrices by normalizing the entries to be unit diagonal and constrained to the range $[-1, 1]$.

We observe that in the ground truth $N \times N$ correlations, magnitude of entries diminish rapidly with distance from main diagonal, indicating weak long-range coupling. This is indicated by the dotted gray line in Figure 4C, which shows the mean absolute value of the entries from the principal diagonal up to the $k$th diagonal band. To focus on meaningful and prominent interactions, we define a *diagonal band* of width $k = 50$ residues ($|i - j| \leq 50$), which captures local and medium-range interactions. This essentially measures residue-residue coupling as a function of *sequence distance* (how distal are $i$ and $j$ along the backbone). We compute the Pearson correlation between predicted and ground-truth residue–residue correlation matrices for the entries along each diagonal band $k$, by iteratively extracting the upper-triangular entries satisfying $|i - j| \leq k$ for $k = 1$ to 50. This is repeated for each of the 82 test set proteins and the median Pearson correlation is reported (each point in Figure 4C). This band-wise analysis enables us to compare the accuracy of coupling signals at increasing residue distances, and we find that DYNAPROT-J (peak correlation of $r = 0.71$) strongly outperforms NMA (peak correlation of $r = 0.59$) particularly at short to mid-range coupling distances, where the coupling is the strongest.

### 4.4 ENSEMBLE GENERATION

As described in Section 3.4, given the output $3 \times 3$ marginal covariances and $N \times N$ residue coupling from DYNAPROT-M and DYNAPROT-J respectively, we reconstruct a full joint covariance using the heuristic defined in Eq. 5. This direct access to the joint distribution enables extremely fast sampling of diverse structures. For evaluation, we sample 250 structures with DYNAPROT, AFMD+TEMPLATES, and NMA to form ensembles for each of the 82 test set proteins in the AFMD split. Following Jing et al. (2024), we assess these ensembles in their flexibility accuracy, distributional similarity, and the ability to reproduce complex observables. For flexibility accuracy, we measure the pairwise RMSD to ground truth MD and RMSF correlation at the global and per-target level. For distributional coverage, we measure the *2-Wasserstein* distance after projecting the ensembles onto the first two principle components derived from the MD trajectory (MD PCA $\mathcal{W}_2$) or the combined (MD+sampled) trajectory (Joint PCA $\mathcal{W}_2$). Table 4 summarizes the ensemble evaluation results across AFMD+TEMPLATES, DYNAPROT, and NMA. DYNAPROT achieves performance

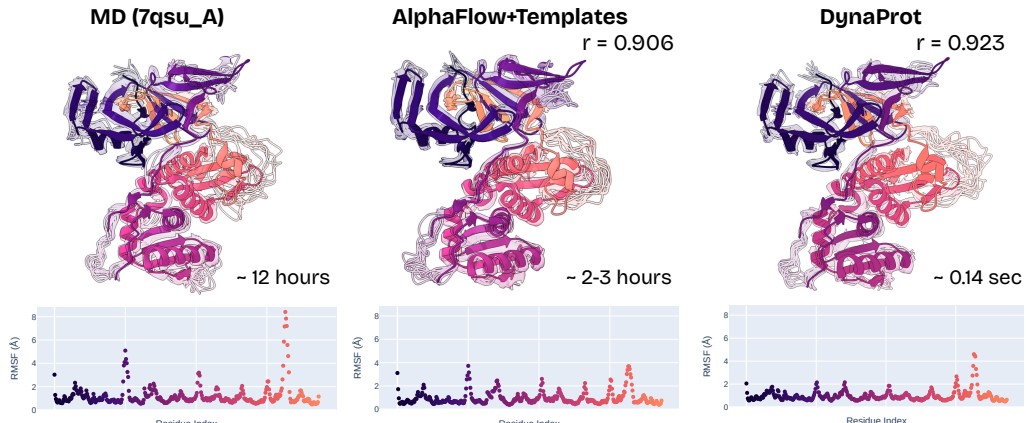

Figure 6: Comparison of DYNAPROT generated ensemble vs. AFMD+T to ATLAS MD simulation (PDB 7qsu_A) overlaid on reference. RMSF Pearson correlation $r$ and sample time reported.

comparable to AFMD+TEMPLATES on key flexibility metrics such as pairwise RMSD and per-target RMSF correlation, while lagging slightly behind on distributional similarity and observable recovery.

Some examples of where DYNAPROT outperforms AFMD+TEMPLATES on ensemble flexibility correlation are visualized in Figure 6 and Appendix A.4. For the visuals, all atom reconstruction is enabled by PULCHRA (Rotkiewicz and Skolnick, 2008). Moreover, DYNAPROT consistently outperforms NMA across nearly all evaluations—except for transient contact prediction—particularly excelling in measures of local flexibility and pairwise distance preservation. Notably, DYNAPROT requires only 2.86 million parameters (vs. 95 million for AFMD+TEMPLATES) and samples ensembles over 70,000× faster on average (∼0.14 s vs. ∼10,000 s), all while

Table 4: Comparison of $C_\alpha$ ensemble evaluation metrics on ATLAS MD Dataset between AFMD+Templates, DYNAPROT, and NMA.

| Metric | AFMD+T | DYNAPROT | NMA |
|---|---|---|---|
| Pairwise RMSD (=2.89) | 2.18 | 2.17 | 0.91 |
| RMSF (=1.48) | 1.17 | 1.10 | 0.52 |
| Global RMSF $r$ ($\uparrow$) | 0.91 | 0.71 | 0.54 |
| Per-target RMSF $r$ ($\uparrow$) | 0.92 | 0.86 | 0.76 |
| MD PCA $\mathcal{W}_2$ ($\downarrow$) | 1.25 | 1.74 | 1.86 |
| Joint PCA $\mathcal{W}_2$ ($\downarrow$) | 1.58 | 2.39 | 2.45 |
| Weak contacts $J$ ($\uparrow$) | 0.62 | 0.51 | 0.43 |
| Transient contacts $J$ ($\uparrow$) | 0.47 | 0.29 | 0.33 |
| # Parameters ($\downarrow$) | 95 M | 2.86 M | – |
| Ensemble sampling time ($\downarrow$) | ∼ 10,000 s | ∼ 0.14 s | ∼ 5.69 s |

being trained only to predict marginal and scalar covariances. This efficiency advantage is maintained when compared against sequence-based methods (i.e. ConfDiff, BioEmu, ESMDiff, see Appendix A.3). DYNAPROT even outperforms them on modeling flexibility and anisotropy.

Finally, we assess DYNAPROT's generalization to longer timescale dynamics, by comparing its zero-shot ensemble of BPTI to the 1ms trajectory from Shaw et al. (2010). Even with these larger conformational changes, DYNAPROT performs reasonably well. It achieves RMSF correlation of 0.88 (c.f. 86 on ATLAS), anisotropy with RMWD of 0.52 Å(c.f. 1.18 Å on ATLAS), and strong recovery of transient contacts ( $J = 0.54$, c.f. 0.29 on ATLAS). See appendix A.3.1 for more.

## 5 CONCLUSION

Protein dynamics is critical for understanding biological function. Existing approaches to modeling dynamics often rely on complex generative models with large-scale PDB pretraining and expensive ensemble generation. In this work, we introduce DYNAPROT, a lightweight and data-driven alternative akin to Normal Mode Analysis (NMA), but designed to directly predict structured dynamics descriptors in the form of per-residue and pairwise Gaussian representations. This formulation enables extreme parameter efficiency while outperforming traditional baselines on key metrics, including flexibility estimation, marginal anisotropy, and residue–residue coupling. Remarkably, DYNAPROT's outputs also support ultra-fast ensemble sampling with reasonable structural fidelity—offering a compelling alternative to conventional ensemble generation methods. While further scaling may be needed to match the full capabilities of state-of-the-art generative methods, our approach highlights a promising alternative grounded in explicitly learning structured representations of dynamics.

## ACKNOWLEDGEMENTS

We would like to thank Alexander Amini, Jeffrey Li, Yuseong (Nick) Oh, Ryan Mei, Manoj Niverthi, Renzo Soatto, and Kabir Doshi for their useful ideas and discussions.

This work was supported by the National Institute of General Medical Sciences of the National Institutes of Health under award number 1R35GM141861-01.

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

## A APPENDIX

### A.1 METHOD DETAILS

**SPD closure of joint reconstruction heuristic (restating Proposition 3.1).** Given marginal covariances $\{\boldsymbol{\Sigma}_{\text{marginal}}^{(i)} \in \mathbb{R}^{3\times3}\}_{i=1}^{N}$ and a correlation matrix $\widetilde{\boldsymbol{C}} \in \mathbb{R}^{N\times N}$ that is symmetric and positive definite, then the reconstructed joint covariance $\boldsymbol{\Sigma}_{\text{joint}} = \boldsymbol{L}_{\text{marginal}} \left(\widetilde{\boldsymbol{C}} \otimes \boldsymbol{I}_3\right) \boldsymbol{L}_{\text{marginal}}^{\top}$ is also symmetric and positive definite.

*Proof.* Let $\boldsymbol{L}_{\text{marginal}} \in \mathbb{R}^{3N\times3N}$ be the block-diagonal matrix defined as

$$\boldsymbol{L}_{\text{marginal}} = \bigoplus_{i=1}^{N} \boldsymbol{L}_i,$$

where each $\boldsymbol{L}_i \in \mathbb{R}^{3\times3}$ is the Cholesky factor (or any valid matrix square root) of the positive definite matrix $\boldsymbol{\Sigma}_{\text{marginal}}^{(i)}$. Since each $\boldsymbol{\Sigma}_{\text{marginal}}^{(i)} \succ 0$, it follows that each $\boldsymbol{L}_i$ is full rank, lower triangular, and has positive diagonal entries. Consequently, $\boldsymbol{L}_{\text{marginal}}$ is full rank and lower triangular with positive diagonal blocks.

Now consider the matrix $\widetilde{\boldsymbol{C}} \otimes \boldsymbol{I}_3 \in \mathbb{R}^{3N\times3N}$. Since $\widetilde{\boldsymbol{C}} \succ 0$ and $\boldsymbol{I}_3 \succ 0$, the Kronecker product $\widetilde{\boldsymbol{C}} \otimes \boldsymbol{I}_3 \succ 0$ as well (Kronecker product of two SPD matrices is also SPD). Finally, the product

$$\boldsymbol{\Sigma}_{\text{joint}} = \boldsymbol{L}_{\text{marginal}} \left(\widetilde{\boldsymbol{C}} \otimes \boldsymbol{I}_3\right) \boldsymbol{L}_{\text{marginal}}^{\top}$$

is a congruence transformation of the SPD matrix $\widetilde{\boldsymbol{C}} \otimes \boldsymbol{I}_3$ by the full-rank matrix $\boldsymbol{L}_{\text{marginal}}$. Since congruence preserves positive definiteness, we conclude:

$$\boldsymbol{\Sigma}_{\text{joint}} \succ 0$$

Moreover, $\boldsymbol{\Sigma}_{\text{joint}}$ is symmetric because it is of the form $\boldsymbol{A}\boldsymbol{B}\boldsymbol{A}^{\top}$. $\qquad\square$

**Multivariate Gaussian Sampling.** Let $\boldsymbol{\epsilon} \sim \mathcal{N}(\boldsymbol{0}, \boldsymbol{I})$ be a standard multivariate normal in $\mathbb{R}^d$, and let $\boldsymbol{\mu} \in \mathbb{R}^d$, $\boldsymbol{\Sigma} \in \mathbb{R}^{d\times d}$ be a symmetric positive definite matrix. Suppose $\boldsymbol{L} \in \mathbb{R}^{d\times d}$ satisfies $\boldsymbol{\Sigma} = \boldsymbol{L}\boldsymbol{L}^{\top}$ (e.g., via Cholesky decomposition or matrix square root). Then the transformation $\boldsymbol{x} = \boldsymbol{\mu} + \boldsymbol{L}\boldsymbol{\epsilon}$ yields a random variable $\boldsymbol{x} \sim \mathcal{N}(\boldsymbol{\mu}, \boldsymbol{\Sigma})$.

*Proof.* Since Gaussian distributions are fully characterized by their first two cumulants (mean and covariance), it suffices to show that the transformed variable has the desired mean and covariance.

Mean of $\boldsymbol{x}$:
$$\mathbb{E}[\boldsymbol{x}] = \mathbb{E}[\boldsymbol{\mu} + \boldsymbol{L}\boldsymbol{\epsilon}] = \boldsymbol{\mu} + \boldsymbol{L} \cdot \mathbb{E}[\boldsymbol{\epsilon}] = \boldsymbol{\mu}$$

Covariance of $\boldsymbol{x}$:
$$\begin{aligned}
\text{Cov}[\boldsymbol{x}] &= \mathbb{E}\left[(\boldsymbol{x} - \boldsymbol{\mu})(\boldsymbol{x} - \boldsymbol{\mu})^{\top}\right] \\
&= \mathbb{E}\left[(\boldsymbol{L}\boldsymbol{\epsilon})(\boldsymbol{L}\boldsymbol{\epsilon})^{\top}\right] \\
&= \mathbb{E}\left[\boldsymbol{L}\boldsymbol{\epsilon}\boldsymbol{\epsilon}^{\top}\boldsymbol{L}^{\top}\right] \\
&= \boldsymbol{L} \cdot \mathbb{E}[\boldsymbol{\epsilon}\boldsymbol{\epsilon}^{\top}] \cdot \boldsymbol{L}^{\top} \\
&= \boldsymbol{L} \cdot \boldsymbol{I}_d \cdot \boldsymbol{L}^{\top} \\
&= \boldsymbol{L}\boldsymbol{L}^{\top} = \boldsymbol{\Sigma}
\end{aligned}$$

$\qquad\square$

A.2   EVALUATION METRICS

**RMWD Variance Contribution.**   To evaluate the efficacy of the marginal Gaussian predictions, we adopt distributional similarity metrics used in Jing et al. (2024). The first of these is the *root mean 2-Wasserstein distance* (RMWD), specifically its variance contribution term. The 2-Wasserstein distance between two multivariate Gaussians has a closed-form expression as follows.

Let $\mathcal{N}_0 = \mathcal{N}(\boldsymbol{\mu}_0, \boldsymbol{\Sigma}_0)$ and $\mathcal{N}_1 = \mathcal{N}(\boldsymbol{\mu}_1, \boldsymbol{\Sigma}_1)$ be two $d$-dimensional Gaussian distributions. The squared 2-Wasserstein distance between them is given by:

$$\mathcal{W}_2^2(\mathcal{N}_0, \mathcal{N}_1) = \|\boldsymbol{\mu}_0 - \boldsymbol{\mu}_1\|_2^2 + \mathrm{Tr}\left(\boldsymbol{\Sigma}_0 + \boldsymbol{\Sigma}_1 - 2\left(\boldsymbol{\Sigma}_1^{1/2}\boldsymbol{\Sigma}_0\boldsymbol{\Sigma}_1^{1/2}\right)^{1/2}\right) \tag{6}$$

This expression consists of two additive components: a mean contribution and a covariance (variance) contribution. This metric is also referred to as the Bures–Wasserstein distance (Bhatia et al., 2019). Since our method predicts only the covariances, we isolate and evaluate only the second term. We define the RMWD variance contribution across $N$ residues as follows:

$$\mathrm{RMWD}_{\mathrm{var}}(\mathcal{N}_0, \mathcal{N}_1) = \sqrt{\frac{1}{N}\sum_{i=1}^{N}\mathrm{Tr}\left(\boldsymbol{\Sigma}_{0,i} + \boldsymbol{\Sigma}_{1,i} - 2\left(\boldsymbol{\Sigma}_{1,i}^{1/2}\boldsymbol{\Sigma}_{0,i}\boldsymbol{\Sigma}_{1,i}^{1/2}\right)^{1/2}\right)} \tag{7}$$

**Symmetric KL Divergence Variance Contribution.**   Alongside the Wasserstein-based metric, we also evaluate the discrepancy between predicted and ground-truth marginal distributions using the *symmetric Kullback–Leibler (KL) divergence*, defined as the mean of the two directed KL divergences mentioned in (Kullback and Leibler, 1951; Jeffreys, 1998):

$$\mathrm{KL}_{\mathrm{sym}}(\mathcal{N}_0 \,\|\, \mathcal{N}_1) = \frac{1}{2}\left[\mathrm{KL}(\mathcal{N}_0 \,\|\, \mathcal{N}_1) + \mathrm{KL}(\mathcal{N}_1 \,\|\, \mathcal{N}_0)\right]$$

For two $d$-dimensional Gaussian distributions $\mathcal{N}_0 = \mathcal{N}(\boldsymbol{\mu}_0, \boldsymbol{\Sigma}_0)$ and $\mathcal{N}_1 = \mathcal{N}(\boldsymbol{\mu}_1, \boldsymbol{\Sigma}_1)$, the KL divergence from $\mathcal{N}_0$ to $\mathcal{N}_1$ is given by:

$$\mathrm{KL}(\mathcal{N}_0 \,\|\, \mathcal{N}_1) = \frac{1}{2}\left[\mathrm{Tr}(\boldsymbol{\Sigma}_1^{-1}\boldsymbol{\Sigma}_0) + (\boldsymbol{\mu}_1 - \boldsymbol{\mu}_0)^{\top}\boldsymbol{\Sigma}_1^{-1}(\boldsymbol{\mu}_1 - \boldsymbol{\mu}_0) - d + \log\frac{\det\boldsymbol{\Sigma}_1}{\det\boldsymbol{\Sigma}_0}\right] \tag{8}$$

This expression consists of both a *mean contribution*—the Mahalanobis term—and a *variance contribution*, comprising the trace and log-determinant terms. Since our method predicts only covariances (and uses the input structure coordinates as means), we isolate the variance terms by omitting ($\mu_1 = \mu_0$) the mean term:

$$\mathrm{KL}_{\mathrm{var}}(\mathcal{N}_0 \,\|\, \mathcal{N}_1) = \frac{1}{2}\left(\mathrm{Tr}(\boldsymbol{\Sigma}_1^{-1}\boldsymbol{\Sigma}_0) - d + \log\frac{\det\boldsymbol{\Sigma}_1}{\det\boldsymbol{\Sigma}_0}\right) \tag{9}$$

To symmetrize the variance contribution of the divergence, we define the symmetric variance KL as:

$$\mathrm{KL}_{\mathrm{symvar}}(\mathcal{N}_0, \mathcal{N}_1) = \frac{1}{2}\left(\mathrm{KL}_{\mathrm{var}}(\mathcal{N}_0 \,\|\, \mathcal{N}_1) + \mathrm{KL}_{\mathrm{var}}(\mathcal{N}_1 \,\|\, \mathcal{N}_0)\right) \tag{10}$$

$$= \frac{1}{4}\left(\mathrm{Tr}(\boldsymbol{\Sigma}_1^{-1}\boldsymbol{\Sigma}_0) + \mathrm{Tr}(\boldsymbol{\Sigma}_0^{-1}\boldsymbol{\Sigma}_1) - 2d\right) \tag{11}$$

### A.3 ADDITIONAL EXPERIMENTS

#### A.3.1 DYNAPROT ZERO-SHOT ENSEMBLE GENERATION OF BPTI

We note that DYNAPROT was trained on the ATLAS MD dataset comprising 100 ns per replicate trajectories. In contrast, D.E. Shaw Research performed simulations of BPTI (PDB: 5PTI) at millisecond-scale revealing structurally distinct conformational states (Shaw et al., 2010). Thus, in an effort to understand DYNAPROT's ability to generalize to long-timescale dynamics, we applied it to BPTI and compared to the DESRES trajectory.

Listed in Table 7b, we compute the ensemble evaluation metrics from Jing et al. (2024) and observe that DYNAPROT performs remarkably well: e.g., RMSF correlation of 0.88 (c.f. 86 on ATLAS), local anisotropy with RMWD of 0.52 Å (c.f. 1.18 Å on ATLAS), and strong recovery of transient contacts (Jaccard similarity 0.54, c.f. 0.29 on ATLAS). These metrics emphasize that DYNAPROT is able to model larger conformational changes at high fidelity.

**MD (5PTI)**          **DynaProt**

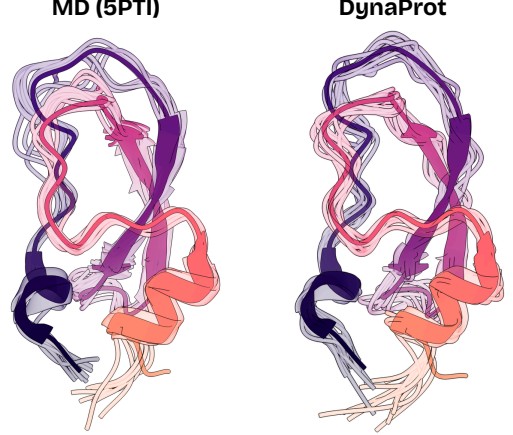

| Metric | DYNAPROT |
|---|---|
| Pairwise RMSD (=1.57) | 1.36 |
| RMSF (=0.84) | 0.86 |
| Per-target RMSF $r$ ($\uparrow$) | 0.88 |
| RMWD Var Contrib ($\downarrow$) | 0.52 |
| MD PCA $\mathcal{W}_2$ ($\downarrow$) | 0.49 |
| Joint PCA $\mathcal{W}_2$ ($\downarrow$) | 0.81 |
| Weak contacts $J$ ($\uparrow$) | 0.54 |
| Transient contacts $J$ ($\uparrow$) | 0.54 |
| # Parameters ($\downarrow$) | 2.86M |
| Ensemble sampling time ($\downarrow$) | $\sim$0.05s |

(b) DYNAPROT zero shot ensemble generation of BPTI (PDB: 5PTI), compared to DESRES MD trajectory (Shaw et al., 2010). Note that the global RMSF Pearson correlation $r$ is omitted as there is only one protein so global = per-target.

(a) Visualization of DYNAPROT zero shot BPTI (PDB: 5PTI) ensemble generation.

#### A.3.2 DYNAPROT ABLATIONS

To test both the importance of DYNAPROT's Riemannian aware loss (log Frobenius norm) and the SE(3) invariance from the IPA layers, we have performed the following ablations listed in Table 5. Unsurprisingly, replacing the log Frobenius norm objective with standard Mean Squared Error loss significantly degrades performance as the

Table 5: DYNAPROT-M ablations of the log Frobenius loss loss and SE(3) invariance.

| Metric | DynaProt | No LogFrob Loss | No SE(3) Invariance |
|---|---|---|---|
| RMWD Var ($\downarrow$) | **1.18** | 2.70 | 1.92 |
| Sym KL Var ($\downarrow$) | **0.91** | 9.26 | 4.46 |
| RMSF $r$ ($\uparrow$) | **0.87** | 0.38 | 0.48 |

optimization is over the space of positive definite covariance matrices, which lies on a well-studied Riemannian manifold. Replacing the IPA blocks with standard MLPs also degrades performance, suggesting that SE(3) invariance is crucial in this low-data, low-parameter regime.

### A.3.3 SEQUENCE INPUT BASELINES

There are many methods worth noting that aim to predict ensembles or dynamics descriptors from sequence itself: standard ALPHAFLOW (AFMD), MSA-SUBSAMPLING, FLEXPERT-SEQ, ESMDIFF (Lu et al., 2024), CONFDIFF (Wang et al., 2024), SEQDANCE (Hou and Shen, 2024), and SEAMOON (Lombard et al., 2024).

Though DYNAPROT's true comparison is NMA as it is a data driven and learnable alternative, we still is we compare against some of these sequence based methods in ensemble generation. DYNAPROT outperforms these methods on local RMSF correlation and marginal anisotropy prediction and is comparable with other distributional metrics. Moreover, the efficiency advantage is clear with DYNAPROT's sub-second sample time.

Table 6: Comparison of DYNAPROT generated with ensemble generation methods that take in sequence as input. ESMDiff, ESM3 entries reported from Lu et al. (2024).

| Metric | DynaProt | ConfDiff OF-r3-MD | AlphaFlow -MD | BioEmu | ESM3 (ID) | ESMDiff (ID) |
|---|---|---|---|---|---|---|
| Pairwise RMSD (=2.89) | 2.17 | 3.43 | **2.89** | 3.57 | - | - |
| RMSF (=1.48) | 1.10 | 2.21 | **1.68** | 2.47 | - | - |
| Global RMSF $r$ ($\uparrow$) | **0.71** | 0.67 | 0.60 | 0.63 | 0.19 | 0.49 |
| Per-target RMSF $r$ ($\uparrow$) | **0.86** | 0.85 | 0.85 | 0.77 | 0.67 | 0.68 |
| RMWD Var Contrib ($\downarrow$) | **1.18** | 1.40 | 1.30 | 2.04 | 4.35 | 3.37 |
| MD PCA $\mathcal{W}_2$ ($\downarrow$) | 1.74 | **1.44** | 1.52 | 2.05 | 2.06 | 2.29 |
| Joint PCA $\mathcal{W}_2$ ($\downarrow$) | 2.39 | **2.25** | **2.25** | 4.22 | 5.97 | 6.32 |
| Weak contacts $J$ ($\uparrow$) | 0.51 | 0.59 | **0.62** | 0.33 | 0.45 | 0.52 |
| Transient contacts $J$ ($\uparrow$) | 0.29 | 0.36 | **0.41** | 0.19 | 0.26 | 0.26 |
| # Parameters ($\downarrow$) | **2.86M** | 12.64M | 95M | 31M | 1.4B | 1.4B |
| Sampling time ($\downarrow$) | $\sim$**0.14s** | $\sim$ 570s | $\sim$ 10,000s | $\sim$ 240s | $\sim$ 70s | $\sim$ 70s |

## A.4 Supplementary Figures

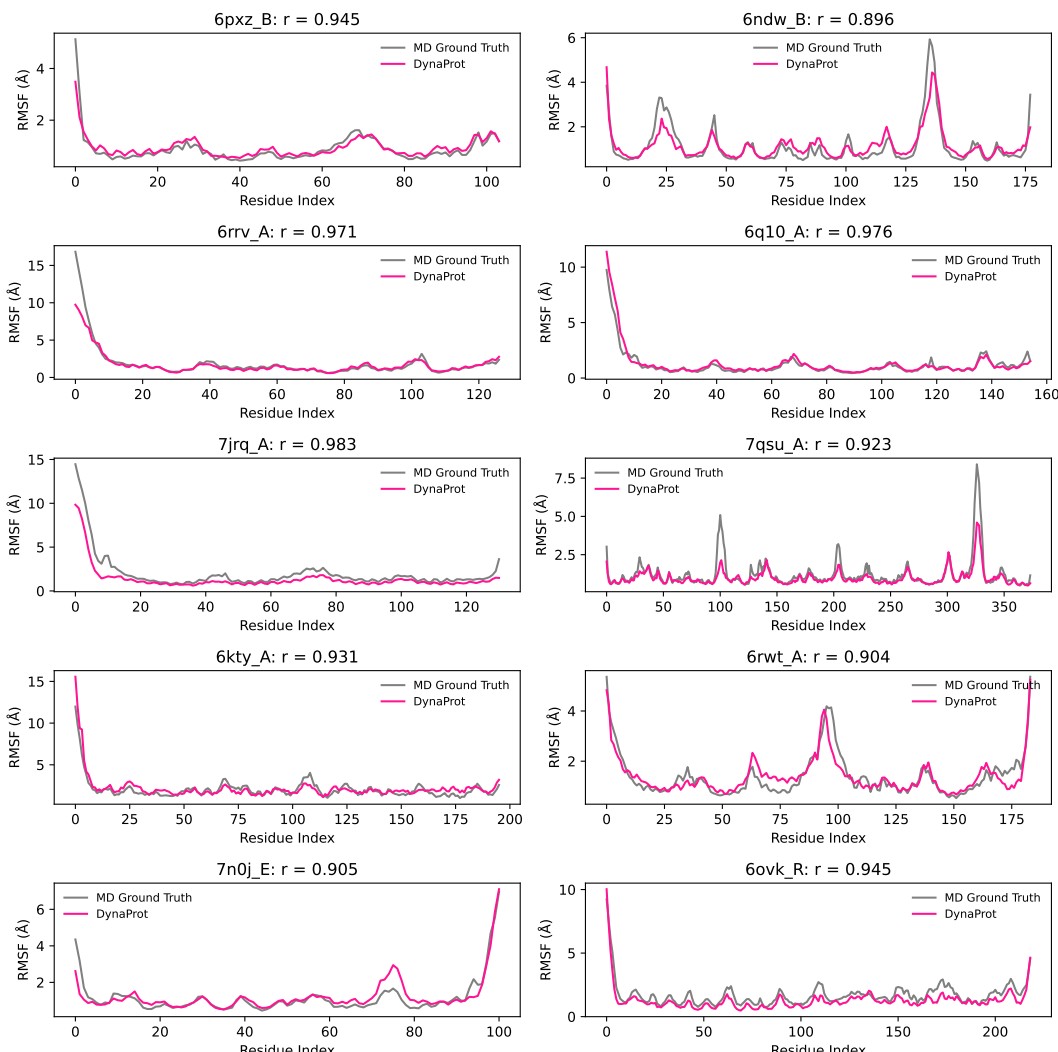

Figure 8: **DYNAPROT-M predicted RMSF correlations.** Visualized test set examples of predicted RMSF per residue (derived from the predicted marginal Gaussians) compared to ground truth RMSF derived from MD trajectories. Pearson correlation coefficient ($r$) between predicted and ground truth RMSF is reported.

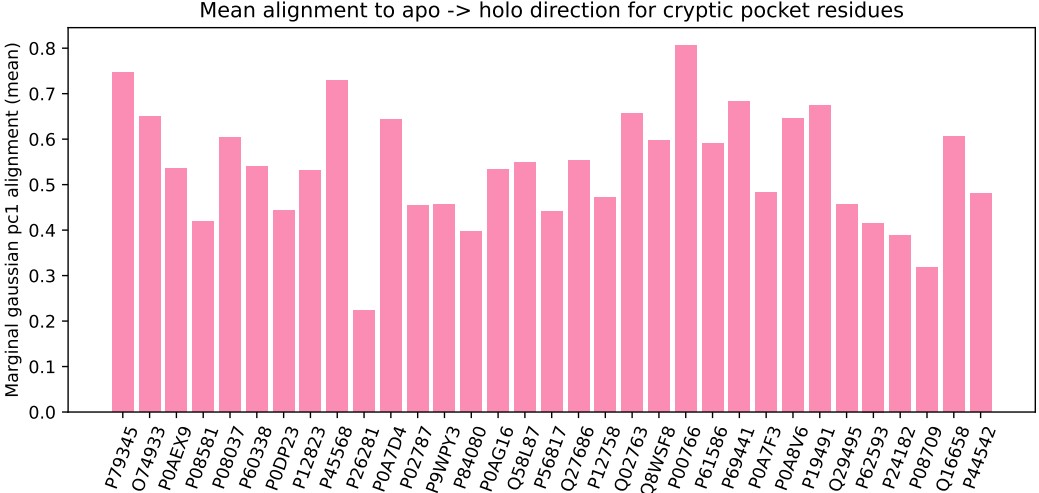

Figure 9: **Evaluation of DynaProt-M zero shot prediction for BioEmu cryptic pocket proteins**. For each reference cryptic pocket protein from Lewis et al. (2024), the apo structure was provided to DynaProt-M, predicting marginal Gaussians for every residue. For each annotated cryptic-pocket residue, a displacement vector from the apo to the holo structure was computed and compared to the principal eigenvector of the respective predicted Gaussian. Cosine similarity between the two vectors quantifies how well DynaProt anticipates the direction of motion associated with pocket formation.

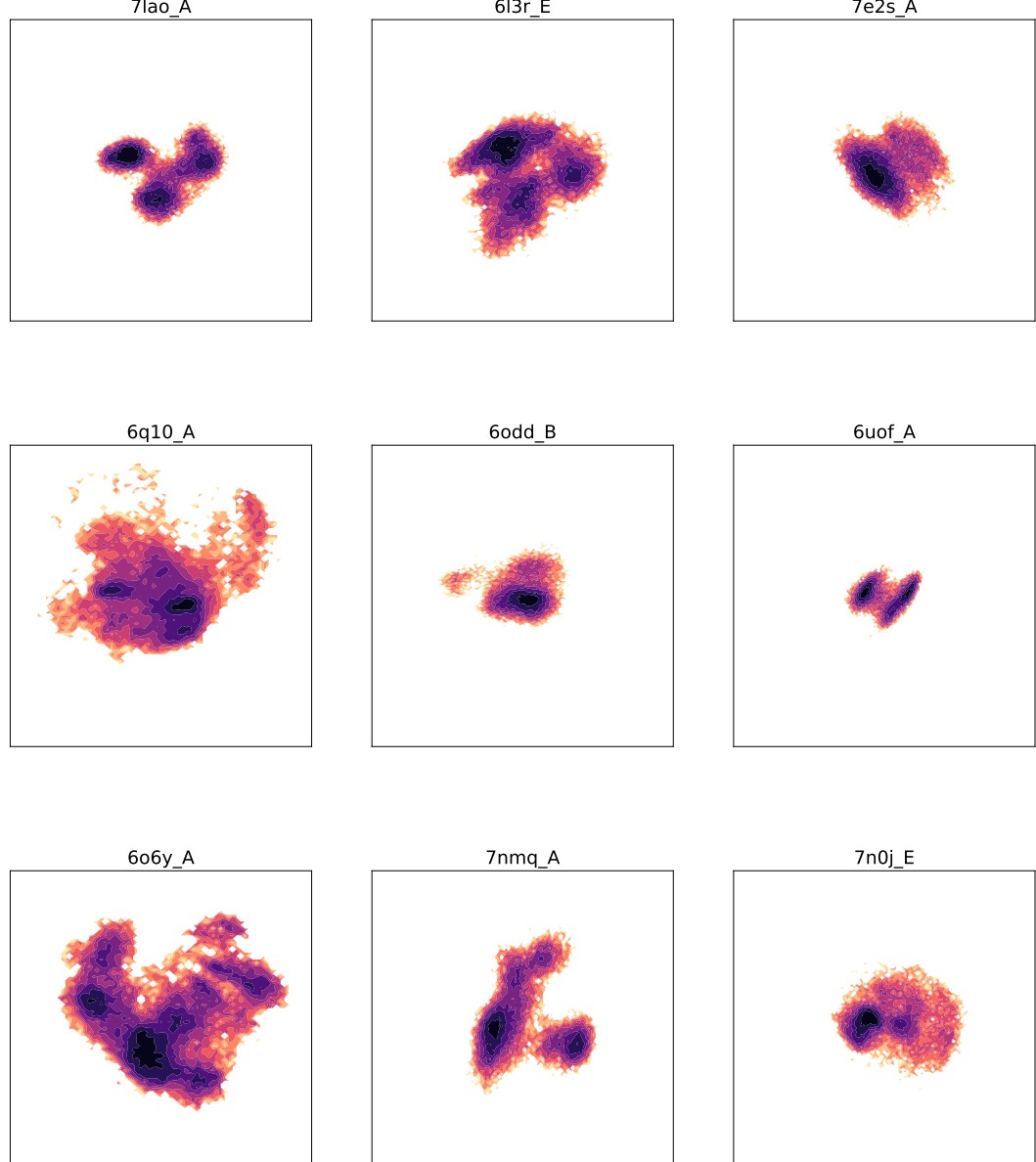

Figure 10: **Sample ATLAS free energy landscapes**. Free energy landscapes obtained by projecting ATLAS MD trajectories onto their top two principal components computed per protein. 2D density estimates transformed into free energy reveal dominant conformational basins explored during simulations.

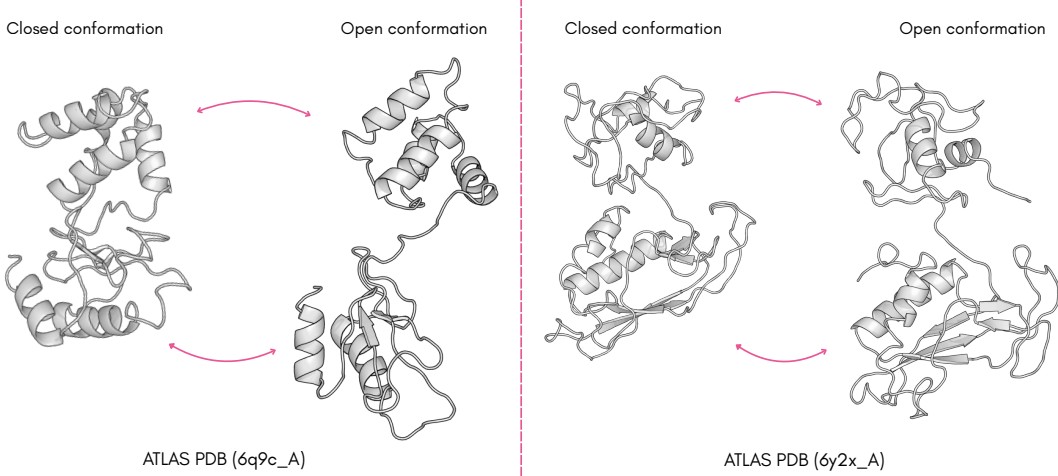

Figure 11: **Sample ATLAS proteins with opening motions.** Visualized are two ATLAS proteins (6q9c_A and 6y2x_A) that display significant opening motions over the course of their trajectories.

