# OpenReview forum: "Learning residue level protein dynamics with multiscale Gaussians"
_ICLR.cc/2026/Conference — ICLR 2026 Poster_

### Official Review · Reviewer_ad6B · 2025-10-19

**Soundness:** 4
**Presentation:** 4
**Contribution:** 3
**Rating:** 8
**Confidence:** 4

**Summary:**

This paper proposes a pipeline for training a model to predict protein dynamics.  The authors model protein dynamics as sampling conformations from a normal distribution with a low-energy mean and a non-identity covariance matrix.  The overall covariance is represented as a combination of a local residue-wise 3×3 covariance matrix and a global residue-wise N×N covariance matrix, matching the dynamics learned from MD simulations.

**Strengths:**

This paper addresses protein dynamics prediction not implicitly through ensemble generation, but directly by modeling the protein’s residue C$\alpha$ covariance matrix.  This approach enables direct analysis of protein dynamics.
Modeling the full covariance matrix using a heuristic that combines the cross-residue covariance matrix with the intra-residue position matrix provides an efficient and meaningful representation of dynamics.
The experiments presented are diverse and interesting, demonstrating the versatility of the proposed approach.

**Weaknesses:**

1. Throughout the paper, the examples presented focus on local dynamics, where the protein does not deviate significantly from the mean. Proteins that undergo large conformational changes may not be well represented by modeling their dynamics as a normal distribution with a single covariance matrix. For proteins that open and close or contain highly flexible regions, the full dynamic range might collapse into a single blurred representation under this parameterization. Nevertheless, this work represents an important step toward explicit modeling of protein dynamics.

2. The authors assume that the provided protein structure is in a low-energy state, but there is no discussion of how the model behaves when the input structure is not in such a state.

**Questions:**

1. When sampling from the normal distribution of protein structures predicted by the model, how often do unphysical protein structures appear?  What prevents this approach from producing impossible torsion angles or bond lengths?

2. How does the model handle proteins with pronounced conformational changes, such as large domain rotations, extensive motion cascades, or highly distinct alternate conformations like those observed in 4OLE?

3. How does the model behave when the provided protein is not in its lowest-energy state?  How would the model perform as part of a dynamical system, where a protein conformation is iteratively updated along the principal directions of each residue? Would the structure return toward the mean state, or would it break down entirely?

---

> ### Author Response · Authors · 2025-11-21
>
> We thank the reviewer for their careful evaluation and positive comments on our method, and we hope to expand on these points in our responses below.
>
> > **Throughout the paper, the examples presented focus on local dynamics, where the protein does not deviate significantly from the mean...Nevertheless, this work represents an important step toward explicit modeling of protein dynamics.**
>
> We appreciate the reviewer's acknowledgement of the significance of our approach. With that said, we wanted to clarify that our evaluations do cover nontrivial conformational motions including domain motions, transient solvent exposure, changes in residue contacts, and clear differentiation of protein order and disorder. To be sure, though, ATLAS simulations are on the shorter end for production simulations and we look forward to applying our training methodology to longer simulation datasets once they become available.
>
> > **When sampling from the normal distribution of protein structures predicted by the model, how often do unphysical protein structures appear?**
>
> The primary output of DynaProt is its marginal Gaussian and coupling predictions. The sampled ensembles are best thought of as a "sketch" of conformational changes derived from the marginal and coupling predictions and are not designed to preserve all physical constraints. With that said, we find that the consecutive Ca distances in these sketches is $3.95 \pm 0.70$ A, on par with the range observed in natural protein structures (~3.8A). For comparison, BioEmu considers distances $\leq 4.5$ A to be valid.
>
> > **How does the model handle proteins with pronounced conformational changes, such as large domain rotations, extensive motion cascades, or highly distinct alternate conformations like those observed in 4OLE?**
>
> We thank the reviewer for their comment. As shown in the newly added Appendix A.4 Fig. 11, the ATLAS dataset does contain proteins that undergo substantial conformational changes, including pronounced hinge like openings. DynaProt’s predicted principal axes often reflect coherent directions of motion (A.4 Fig 8), and we would expect these axes to align with the dominant directions of large scale conformational changes when such motions are present.
>
> >**How does the model behave when the provided protein is not in its lowest-energy state?**
>
> To improve generalizability and reduce template bias, we train DynaProt with data augmentations by randomly sampling input frames from the ATLAS trajectory. Hence, input structures not at the lowest energy state should not pose any issue (as long as they are plausible frames from an MD trajectory).
>
> >**How would the model perform as part of a dynamical system, where a protein conformation is iteratively updated along the principal directions of each residue? Would the structure return toward the mean state, or would it break down entirely?**
>
> In the current version of the architecture, this successive ensemble sampling is not actually possible, as the outputs only involve translational motions of the C-alpha, whereas the input requires at least the backbone frame orientations. Hence, while we can compute evaluations based on C-alpha positions, we cannot feed sampled structures back to the model. Nevertheless, the reviewer suggests an interesting direction and we look forward to trying it out in future work!

---

> > ### Comment · Reviewer_ad6B · 2025-11-25
> >
> > Thank you for the rebuttal. I will maintain my original score.

---

### Official Review · Reviewer_LoLM · 2025-10-27

**Soundness:** 2
**Presentation:** 3
**Contribution:** 2
**Rating:** 4
**Confidence:** 4

**Summary:**

This paper introduces DYNAPROT, a novel, lightweight, SE(3)-invariant deep learning framework for directly predicting protein dynamics from static protein structures. Unlike molecular dynamics (MD) simulations, which demand substantial computational resources, or generative models requiring extensive pre-training and costly sampling procedures, DYNAPROT is an efficient alternative.

**Strengths:**

Compared to models such as DYNAPROT (with 2.86 million parameters in total) and AlphaFlow (95 million parameters), the parameter count is reduced by several orders of magnitude. Its inference and sampling speeds far exceed those of other methods, rendering it exceptionally practical. Moreover, the model performs well and is trained solely on molecular dynamics data from approximately 1,000 proteins.

**Weaknesses:**

The model was trained on the ATLAS dataset comprising 100 ns simulations. For numerous biologically significant processes—such as large-scale domain motions and allosteric transitions—100 ns is wholly inadequate, as these typically unfold over timescales ranging from microseconds to milliseconds or longer. Consequently, the model may have predominantly learned local, transient thermal fluctuations while exhibiting limited capacity to capture large-scale conformational changes. Although the paper's zero-shot testing results for BPTI appear favourable, this represents a single protein case study. One cannot conclude from this solitary example that the model generalises well to all long-timescale dynamics.

**Questions:**

The model was trained on 100ns of data. Beyond the single example of BPTI, have the authors tested the model on additional proteins exhibiting known long-timescale (e.g. > 1us) dynamic characteristics? Is there a possibility that the model might overfit to short-timescale fluctuations, thereby underestimating genuine large-scale motions?

---

> ### Author Response · Authors · 2025-11-21
>
> We thank the reviewer for their thoughtful assessment and acknowledge the main concern is generalization to long timescale dynamics, which we address below.
>
> ### **1. Concern about ATLAS Dataset**
>
> We acknowledge that 300 ns per system is on the low end for production simulation timescales. With that said, 300 ns can cover a surprising amount of conformational motion, including domain decouplings, secondary structure changes, and clear differentiation between order and disorder; the conformational diversity certainly goes beyond local mode exploration (Appendix A.4 Fig. 10 and 11). While not perfect, ATLAS is a robust, well-curated dataset with nontrivial motions and crucially focuses on the key axis of generalization that longer-timescale simulation datasets do not. Its role may be compared to that of CIFAR-10 --- a simplified evaluation setting to be sure, but a highly useful sandbox for algorithmic development, which is the focus of this work.
>
> ### **2. Value beyond ATLAS**
>
> Despite being trained on relatively short simulations, DynaProt already demonstrates strong practical performance compared to methods which are designed to be timescale-agnostic. It substantially outperforms classical methods like Normal Mode Analysis, and even surpasses generative models such as BioEmu (Appendix A.3.3 Table 6), despite BioEmu being trained on far longer internal simulation datasets—and DynaProt achieves this with markedly lower inference time (cf. 0.14s vs. 240s). Its ability to generalize zero shot to BPTI (far outside ATLAS timescale) already shows that the model can capture more than short timescale thermal noise. And, we believe its marginal Gaussian predictions are directly actionable in structural biology workflows: in our adenylosuccinate synthase cryptic-pocket discovery experiment, the predicted marginals correctly highlighted long-timescale pocket forming motions (Fig. 5), which are from experimental structures.
>
> Nevertheless, we acknowledge the reviewer's concern about ATLAS-focused evaluations, so to better quantify DynaProt's generalizability, we have compiled an additional experiment below.
>
> ### **3. New experiment on DynaProt zero-shot generation against NMR ensembles**
>
> > Beyond the single example of BPTI, have the authors tested the model on additional proteins exhibiting known long-timescale (e.g. > 1us) dynamic characteristics? Is there a possibility that the model might overfit to short-timescale fluctuations, thereby underestimating genuine large-scale motions?
>
> Unfortunately, large scale datasets of long MD simulations do not exist publicly. To assess DynaProt's ability to cover motions of arbitrary timescale in the absence of such MD data, we compared DynaProt predictions to experimentally resolved **NMR structural ensembles** from the PDB that by definition **capture the full conformational heterogeneity of the protein (including long timescale dynamics)**, well beyond thermal fluctuations.
>
> We curated the top 100 largest non redundant NMR resolved proteins (up to 500 aa) in the PDB. NMR ensembles with only 1 model were discarded. The final set (listed below for your reference) includes proteins with known conformational variability, including IDRs.
>
> NMR PDB IDs: [2lgc, 5nmy, 2kdy, 2rsv, 2p01, 6bqi, 5gjj, 2k6h, 6amv, 2d21, 1ezo, 2lgz, 2ayx, 6hd2, 2jqy, 2kdo, 7mzw, 1z87, 2mbg, 6f3k, 2roq, 1tvc, 6kvg, 2jtc, 2jz4, 2mhk, 1ah2, 2qmv, 6i3r, 7bbb, 7ry6, 2myi, 7dme]
>
> For each NMR entry above, we ran DynaProt inference (zero shot) on the first listed PDB model to generate ensembles. We compared these predicted ensembles to the NMR ensembles and obtained the following results:
>
> | Metric               | DynaProt zero shot  (NMR) |
> |----------------------|-----------------|
> | RMSF (=2.61)         | **1.26**        |
> | Pairwise RMSD r (↑)  | **0.87**        |
> | Global RMSF r   (↑)  | **0.69**        |
> | Per-target RMSF r (↑)| **0.82**        |
> | Weak contacts J (↑)  | **0.44**        |
> | Transient contacts J (↑) | **0.29**        |
>
> These results are comparable to DynaProt's performance on ATLAS, despite no retraining. With this NMR based evaluation and the BPTI evaluation, we have demonstrated that DynaProt can model conformational changes beyond short timescales even on proteins that have IDRs. Combined with its strong performance relative to NMA and its demonstrated utility in cryptic pocket discovery, we hope this addresses the reviewer's concerns regarding pretraining on ATLAS.
>
>
>
> With these additional NMR experiments, our clarifications regarding ATLAS, and DynaProt’s demonstrated performance beyond short-timescale fluctuations, we hope we have fully addressed the reviewer’s concerns and would be grateful if they would consider raising their score.

---

### Official Review · Reviewer_HMqK · 2025-10-27

**Soundness:** 3
**Presentation:** 3
**Contribution:** 2
**Rating:** 2
**Confidence:** 4

**Summary:**

The paper presents DYNAPROT, an innovative and lightweight SE(3)-invariant model that predicts protein dynamical features—marginal anisotropy and residue-level couplings—directly from static structures. It achieves accuracy comparable to leading generative models without large-scale PDB pretraining, while being more efficient. By constructing covariance matrices and introducing scalar covariances, the model can capture some key features, which are useful for fast ensemble generation.

**Strengths:**

1. The paper is well organized and clearly stated.
2. The authors propose a protein dynamics framework based on a Gaussian hierarchical representation, which incorporates per-residue covariance tensors and the global coupling matrix into a Gaussian hierarchy.
3. Compared with a single scalar RMSF, the covariance matrix is richer: it captures both the shape and direction of fluctuations.

**Weaknesses:**

1. The core issue is model misspecification. A Gaussian hierarchy simplifies learning and sampling, and it can not reflect the thermodynamics of MD data. Molecular energy landscapes are multi-well with distinct basins. Gaussian models are locally valid only near a single basin (approximately harmonic vibrations) and cannot capture complex, anharmonic shapes or multimodality across multiple minima. This defect was masked in the atlas dataset. Proteins in Atlas are relatively large, so the existing MD trajectories predominantly capture local fluctuations rather than informative transitions. Could the authors validate the method on simpler systems—e.g., a 2D double-well potential and fast-folding proteins—to demonstrate effectiveness of model?
2. Although Eq. 5 and Proposition 3.1 guarantee positive semidefiniteness, the construction in Eq. 5 lacks clear physical motivation. The marginal covariance $\Sigma_{marginal}$ rightly emphasizes per-residue anisotropy, which is a strength. However, the coupling matrix $C$ enforces isotropic interactions—i.e., identical influence along all three spatial directions—between residues. This design oversimplifies residue–residue coupling and risks weakening key directional effects that are critical in complex biomolecular systems. Could the authors elaborate on the motivation and theoretical basis for this design choice?
3. When constructing $C$, the use of **mean pooling** as a scalar projection to compute each $C_{ij}$ appears casual and overly simplistic for modeling residue–residue coupling. A more completed ablation is needed to justify this design choice.
4. What is the $\mu$ of the BPTI results in Appendix A.3.1? Are the two main conformational changes caused by $\mu$ or by $\Sigma$?

**Questions:**

The same as weaknesses. In multi-conformation generation tasks, more attention is paid to the changes in system transitions, (rather than the local fluctuation). If the author is willing to explain this point, I am willing to increase the score

---

> ### Author Response · Authors · 2025-11-21
>
> We appreciate the reviewer's comments and the opportunity to clarify our contribution. To address your primary concern, we would like to re-articulate the relationship between DynaProt, ensemble generation, and normal mode analysis.
>
> ### **Model misspecification**
>
> Although it is true that the ground truth ensemble is not Gaussian, Gaussian representations can serve as informative, parsimonious indicators of conformational dynamics (in the same way that RMSF profiles can do so). Indeed, the concept of a Gaussian representation of dynamics is not new---it derives from normal mode analysis (NMA), which models conformational ensembles with harmonic potentials (and thus Gaussian distributions). NMA is a well validated biophysical modeling technique used in many studies [1][2][3][4], even though it is technically also misspecified.
>
> As we motivated in the introduction, the primary intent of DynaProt is to improve upon NMA by learning the parameters of the Gaussian from data, rather than simplified physical assumptions. Despite the availability of MD and generative models, NMA remains widely used because it provides fast, interpretable insight into how a structure is likely to move. **This suggests that improving upon the accuracy of NMA—while retaining its speed and simplified modeling assumptions—can make a material difference for structural biology.**
>
> From another perspective, the broader role of techniques with simplified distribution assumptions (DynaProt, NMA) is to quickly preview or "sketch" dynamics. At this task, which is evidently valuable, DynaProt outperforms NMA across the board. We can imagine structural biologists using it as a PyMOL plugin in which dynamic "Gaussian blobs" are immediately overlaid on the loaded structure, revealing biologically meaningful insights (e.g. cryptic pocket formation, Section 4.2.1). Samples from the DynaProt ensemble can also be generated at minor additional cost, and can be followed up with full distributional sampling (generative models, MD) if desired.
>
> ### **ATLAS coverage of multiple modes**
>
> >Gaussian models are locally valid only near a single basin (approximately harmonic vibrations) and cannot capture complex, anharmonic shapes or multimodality across multiple minima. This defect was masked in the atlas dataset.
>
> Although Gaussians are a restrictive distribution family, their covariance matrices are capable of representing axes of motion corresponding to mode hopping in the underlying ensemble. Indeed, the ATLAS dataset contains nontrivial mode-hopping motions. To illustrate this point, in Appendix A.4 Figure 10, we plot the free energy landscapes of several systems from ATLAS of varying sizes. These examples illustrate that the dataset does not consist solely of small harmonic fluctuations; rather, many systems exhibit meaningful mode-hopping and large conformational motions (Appendix A.4 Figure 11). We emphasize that DynaProt’s predicted covariance matrices capture these directions with significantly higher accuracy than NMA, the only previous comparable method (e.g. PC W2 of 1.74).
>
> Additionally, our DESRES BPTI experiment (Appendix A.3.1) demonstrates that our model can capture motions from simulations substantially longer than our training trajectories and which exhibit nontrivial conformational rearrangements.
>
> ### **Simpler systems**
> >simpler systems—e.g., a 2D double-well potential and fast-folding proteins
>
> We appreciate the suggestion to evaluate on simpler systems. However, we do not believe validation on fast-folding proteins is appropriate because those are simulated at artificially elevated temperatures to induce unfolding, whereas DynaProt is trained at physiological temperatures.
>
> ### **Motivation of the coupling matrix**
>
> The reviewer makes an insightful observation about the coupling matrix. It is true that there are nine degrees of freedom for reach residue-residue pair, even within the Gaussian model class. Earlier in development, we trained models to predict all nine of these quantities, but found it challenging to enforce positive-definiteness (e.g., by predicting correlations instead of covariances) while retaining stable training. We hypothesized that this could be data related: with only ~1400 trajectories, directly learning all nine cross anisotropy parameters per residue pair is unnecessary, as the empirical estimates of these quantities would be noisy at this sample size.
>
> Our coupling matrix construction represents a slight restriction in the model class; nevertheless, it is sufficient for significantly improved results over NMA (which we note has its own set of restrictions in model class). Importantly, this formulation still preserves the full per-residue anisotropy in the marginals, which NMA does not accurately account for.

---

> > ### Author Response · Authors · 2025-11-21
> >
> > ### **Mean pooling**
> >
> > Our goal in constructing the labels for residue-residue coupling $C$ was to obtain a scalar coupling strength $C_{ij}$ for each residue pair that can be interpreted as a correlation weight and safely used in a global SPD matrix. This requires a symmetric, permutation-invariant projection from the pairwise 3 by 3 covariance to a single number. Early in development, we explored other reductions (e.g. trace) and found comparable coupling results. We ended up choosing mean pooling because of its numerical stability and unlike the trace it captures _some_ information about residue $i,j$ cross anisotropy by averaging all the entries in the block rather than only the diagonal. Since our aim was to define a robust coupling mechanism that shows substantially improved performance over NMA, we opted for the simplest aggregation.
> >
> >
> > ### **Clarifications about BPTI**
> >
> > We thank the reviewer for their question, and apologies if we have misunderstood. If the question concerns whether we use different means $\mu$ as input to DynaProt, then this is not the case. For all our experiments, including BPTI, we treat the mean $\mu$ as a single input structure (PDB 5PTI in the case of BPTI) and DynaProt predicts the covariance $\Sigma$. So, all conformational variability in our generated ensembles arise entirely from the single learned covariance $\Sigma$.
> >
> > We thank the reviewer as this raises an interesting direction for future work: if multiple low energy structures were known/predicted a priori, one could in principle generate a Gaussian around each and combine them into a simple mixture model to represent a richer landscape.
> >
> >
> > ### **Questions**
> >
> > >In multi-conformation generation tasks, more attention is paid to the changes in system transitions, (rather than the local fluctuation).
> >
> > Our assessments of ensemble generation quality do include metrics that attend to system transitions - principle axes of motion in the (multimodal) ground truth ensemble and rare events such as weak or transient residue-residue contacts. All of our distributional metrics (RMWD, Wasserstein distances) are measured against the system transitions in the multimodal ground truth ensemble, not local fluctuations.
> >
> >
> > Overall, we greatly appreciate the reviewer’s insightful comments, which have helped us further clarify the scope and motivation of our work, specifically that DynaProt is positioned as an improved alternative to NMA. We hope that the explanations and additional analyses resolve any concerns, and we would be grateful if the reviewer would consider raising the score in light of these clarifications.
> >
> > ---
> > [1] Yamato, Takahisa, and Olivier Laprévote. "Normal mode analysis and beyond." Biophysics and Physicobiology 16 (2019): 322-327.
> >
> > [2] Kolossváry, István. "A Fresh Look at the Normal Mode Analysis of Proteins: Introducing Allosteric Co-Vibrational Modes." JACS Au 4.4 (2024): 1303-1309.
> >
> > [3] Bahar, Ivet, et al. "Normal mode analysis of biomolecular structures: functional mechanisms of membrane proteins." Chemical reviews 110.3 (2010): 1463-1497.
> >
> > [4] Miller, David W., and David A. Agard. "Enzyme specificity under dynamic control: A normal mode analysis of α-lytic protease." Journal of molecular biology 286.1 (1999): 267-278.

---

> ### Comment · Reviewer_HMqK · 2025-11-24
> **The last question**
>
> For conformations with relative large variance or sharp structural changes, how do you ensure the generated structures remain physically valid? Naive Gaussian sampling can easily introduce steric clashes and bond-length violations. Could you show the generation results for the 6q9c A and 6y2x examples mentioned in Figure 11?
>
> The paper builds upon the concept of NMA. However, NMA is inherently better suited for capturing changes in local regions, making it highly dependent on the input PDB. MSF typically serves as an evaluation metric for diversity, (rather than generation pipeline), so one can expect that when the variance is excessive, the quality of samples generated from a Gaussian distribution will inevitably degrade.
>
> While the model may outperform traditional methods, it does not solve the fundamental problem.

---

> > ### Comment · Reviewer_HMqK · 2025-11-24
> >
> > I thank the authors for their response. First, I acknowledge that the paper has excellent presentation. The overall structure is very clear.
> >
> > However, I maintain my original score. The authors have not addressed my main concern.
> >
> > - I remain skeptical about the rationality of using a network to learn the covariance matrix. I stand by the view that harmonic-based models are fundamentally unsuitable for systems with high energy barriers and large variations. This is of limited benefit to the ML generation task, (although I understand that this paper aims to address different problems and operates with different computational costs).
> > - While the design of the covariance matrix is mathematically valid (i.e., symmetric and positive definite), the paper still lacks an explanation of the underlying principles and a sufficient justification for this design choice.

---

> > > ### Author Response · Authors · 2025-11-24
> > >
> > > Thanks for the quick response and continued engagement! We hope to continue the discussion as we think there may be some misunderstanding about the intended scope of DynaProt.
> > >
> > > DynaProt is not designed to be an ensemble generation method, and we do not position it as such. Its primary contribution is the ability to predict fast, residue level anisotropic dynamics readouts---directional information about how the residues move. One can envision a PyMOL plugin (analogous to existing NMA plugins) that draws arrows and Gaussian "blobs" on an input structure to show directions of likely motion under MD simulations. From the widespread use of NMA, this kind of prediction is of clear utility, and DynaProt is the first deep learning method for predicting such readouts. As demonstrated in our cryptic pocket experiment (Section 4.2.1) and additional tests on the BioEmu benchmark (Appendix A.4 Figure 9), DynaProt's readouts deliver valuable insights, without any ensemble generation.
> > >
> > > To further visualize the predicted motions, DynaProt can optionally produce a lightweight ensemble "sketch" at minor additional cost, providing users a quick sense of the dominant fluctuations of the protein. This sketch does not guarantee physical constraints, but conveys information about residue-residue couplings. We do not claim that the sketch represent the true ensemble - which is not Gaussian and could not be reproduced by DynaProt, even if learned perfectly. However, we use established metrics to show that the information content of the ensemble sketches is comparable to the outputs of much heavier methods. We apologize if this gave the impression that DynaProt is intended to be an ensemble generation method itself.
> > >
> > > Finally, the value of DynaProt is not merely in the current model artifact, which is necessarily limited by extant training data, but also the algorithmic framing of the learning problem (which can be applied to new training data available in the future). We are the first to propose a deep learning formulation that exactly mirrors and improves upon the dynamics readouts of NMA. We argue this is of concrete value to the protein ML community. Respectfully, we think that "solving the fundamental problem" is of course the end goal, but not a necessary condition for a contribution to be meaningful.
> > >
> > > We hope this clarifies our position and look forward to better understanding your concerns about DynaProt's merits as a protein ML contribution.

---

> > > > ### Comment · Reviewer_HMqK · 2025-11-25
> > > >
> > > > I thank the authors for their response. In light of the authors' sincere engagement and the paper's excellent presentation, I am willing to raise my score to 4. However, I maintain that the work still suffers from the following two major issues, which I leave for the AC to evaluate.
> > > > - Fundamental Rationality: I remain skeptical about the rationality of using a network to learn the covariance matrix. I stand by the view that harmonic-based models are fundamentally unsuitable for systems with high energy barriers and large variations.   Authors fail to explain this limitation.
> > > > - While the design of the covariance matrix is mathematically valid (i.e., symmetric and positive definite), the paper still lacks an explanation of the underlying principles and a sufficient justification for this design choice. This point remains insufficiently explained. (Also modeling RMSF appears weird for me, although it is a metric for local diversity).

---

> > > > > ### Author Response · Authors · 2025-12-01
> > > > >
> > > > > We greatly appreciate that the reviewer raised their score to a 4 and their positive comments about the paper's presentation. Although the discussion has concluded, we want to briefly state our perspective for the ACs given the extended back and forth on this point.
> > > > >
> > > > > Regarding the Gaussian (harmonic) approximation, we remain of the view expressed earlier: it has long been proven useful in practice through NMA, which relies on the same formulation and has produced many biological insights. DynaProt is intended in this same role as a fast anisotropic dynamics readout that significantly improves upon NMA.
> > > > >
> > > > > Our full joint covariance construction is simply the multivariate extension of the univariate case: each residue’s learned anisotropy (the Cholesky factor of its marginal covariance) is combined across residues and scaled by the learned correlation $c_{i,j}$. As for RMSF, it is a widely used scalar flexibility measure and not at all unusual to model; DynaProt recovers it implicitly while also providing the additional directional and anisotropic information that RMSF alone cannot capture.
> > > > >
> > > > > We believe DynaProt provides concrete value to the protein ML community. The predicted marginals already show practical utility for cryptic pocket identification (Sec 4.2.1 ; A.4 Fig.9) and the generated ensemble sketches are informative even in challenging OOD cases such as BPTI (A.3.1).

---

### Official Review · Reviewer_6FkX · 2025-11-01

**Soundness:** 3
**Presentation:** 4
**Contribution:** 3
**Rating:** 6
**Confidence:** 3

**Summary:**

DynaProt is a Gaussian-based framework for ultra-fast protein conformation ensemble generation. Unlike traditional generative models that learn protein structure distributions via sampling, DynaProt learns a hierarchical Gaussian parameterization composed of residue-level anisotropic Gaussians and a protein-level isotropic residue–residue covariance matrix that captures coupling between residues.
This formulation enables DynaProt to analytically compute dynamics-related quantities such as RMSF without sampling; reconstruct the full 3N×3N covariance matrix using simple heuristics to support conformation sampling, generate large conformational ensembles extremely efficiently by sampling from reconstructed Gaussian distribution.
Compared with classical normal mode analysis, which also relies on Gaussian approximations, DynaProt learns variances and couplings directly from data rather than through simplified physical potentials, resulting in more accurate ensemble statistics.
Experimental results show that DynaProt achieves competitive performance on some ensemble-level metrics compared to diffusion generative models while being orders of magnitude faster. Although some accuracy is sacrificed due to the inherent Gaussian assumptions, the method offers a compelling balance between speed, accuracy, and physical interpretability, making it a strong and distinctive solution for conformation sampling and protein dynamics analysis.

**Strengths:**

- The manuscript is well organized, clearly written, and easy to follow, with clean and informative figures that effectively support the main ideas.
- The model addresses a unique and practical perspective in conformational modeling, where not only accuracy but also the ability to efficiently generate large numbers of samples is essential. DynaProt improves upon classical normal mode analysis while being orders of magnitude faster than modern generative models.
- The method produces versatile outputs (e.g., RMSF, residue-level anisotropic Gaussian profiles, residue-residue coupling) and achieves competitive accuracy on some ensemble-level metrics compared to current state-of-the-art models.

**Weaknesses:**

- The core limitation lies in the Gaussian approximation and reliance on input structures, which restricts expressiveness compared to more flexible generative models - but this is aligned with the paper’s goal of trading some accuracy for major gains in speed and efficiency.
- Given this design philosophy, clearly demonstrating the cases where DynaProt is useful becomes important. Cryptic pocket modeling is a relevant application, but only one case study is provided.
- The current formulation models only coarse-grained residue-level dynamics. It would strengthen the paper if the authors could demonstrate potential extensions to atomic-resolution modeling or protein-ligand complexes.

**Questions:**

### 1. Relationship between DynaProt-M, DynaProt-J, and DynaProt

In the Experiments “preprocessing” section, DynaProt-M, DynaProt-J, and the IPA backbone appear to be trained as three separate models. However, in Section 4.4, the final DynaProt results are described as being composed from the output of DynaProt-M and  DynaProt-J. Could the authors clarify which is the case? And if separate models are trained, why is it necessary (vs. a single model with two prediction heads)?

### 2. Cryptic pocket case study and sampling
In Section 4.2.1, the authors show that predicted marginal variances can indicate ligand-binding dynamics, such as cryptic pocket formation. Have the authors sampled from the full reconstructed Gaussian distribution for this protein, and verified whether both open and closed conformations can actually be generated?

Additionally, BioEmu (https://www.science.org/doi/10.1126/science.adv9817) includes several benchmarks with cryptic pocket motions. Evaluating DynaProt on those systems and reporting pocket coverage metrics would further strengthen the results.

### 3. Extending to atomic-level or protein-ligand complexes

DynaProt demonstrates impressive speed and potential scalability for residue-level modeling. Is it feasible to extend this approach to atomic-level resolution or to explicitly model protein-ligand complexes? Are there technical challenges such as dimensionality of the covariance matrix or parameterizing atom-level couplings?

---

> ### Author Response · Authors · 2025-11-21
>
> We thank the reviewer for their constructive feedback and provide clarifications to each point below.
>
> ### **Relationship between DynaProt-M, DynaProt-J, and DynaProt**
>
> To clarify, there are only two models: DynaProt-M has an IPA backbone with a marginal Gaussian head, while DynaProt-J has a IPA backbone with the pairwise coupling head. The full joint is recovered by composing the outputs from these two models. We train them separately because the two tasks have different practical use cases: marginal Gaussians do not require pairwise features and keeping DynaProt-M standalone makes it even more stable and lightweight (< 1M params). For example, we imagine DynaProt-M could be utilized as a PyMol plugin where the marginal gaussian blobs could be predicted and overlaid almost instantly upon loading in a structure. You are right that single multi head model is certainly feasible, but we chose the separation mainly to reduce unnecessary computation when only marginal predictions are desired.
>
>
> ### **Synthetase cryptic pocket full sampling**
>
> > Have the authors sampled from the full reconstructed Gaussian distribution for this protein, and verified whether both open and closed conformations can actually be generated?
>
> In Section 4.2.1, our goal was to demonstrate the utility of DynaProt's marginal Gaussian predictions in isolation, which already provide actionable information revealing the principal residue directions of motion associated with the cryptic pocket formation. This is why ensembles were not generated for that experiment.
>
> That said, the reviewer brings up a valuable point to sanity check whether the ensembles also recover the holo form. To test this, we provided the apo structure (1ADE_A) as input to DynaProt and zero shot sampled an ensemble of 1000 structures calculating RMSD to the holo form (1CIB_A) for each. We found that approximately 1.3% of samples could be classifed as holo, using a 1.5 angstrom cutoff (same as BioEmu experiments). We note that BioEmu considers 0.1% coverage as successful recovery of the holo state.
>
>
> ### **New cryptic pocket experiment from BioEmu protein list**
> >Evaluating DynaProt on those [BioEmu cryptic pocket benchmark] systems ... would further strengthen the results.
>
> We thank the reviewer for this helpful suggestion. BioEmu does indeed contain several relevant cryptic pocket systems, but unfortunately the repository is not compatible with C-alpha only ensembles. Hence, we developed our own experiment to test DynaProt's strength on the curated apo/holo PDB pairs. For each pair, we provide the apo structure as input to DynaProt-M and predict the marginal Gaussians. For each pocket residue, we computed the direction vector from apo $\rightarrow$ holo and compared it to DynaProt's top eigenvector at that residue using cosine similarity. As shown in Appendix A.4 Fig. 9, the mean alignment scores are high across the benchmark (0.5--0.8 mean cosine similarity for most proteins), indicating that DynaProt's predicted Gaussians have the same directionality as the experimentally observed pocket formation motion.
>
> We appreciate this suggestion, as it highlights one of DynaProt's core advantages: the marginal Gaussians capture residue level anisotropy and directionality, which cannot be obtained by isotropic or scalar predictors like RMSF. As we see here, these directional signals can provide actionable structural insight and take less than a second to predict.
>
> ### **Extending to atomic-level or protein-ligand complexes**
> >  Is it feasible to extend this approach to atomic-level resolution or to explicitly model protein-ligand complexes? Are there technical challenges such as dimensionality of the covariance matrix or parameterizing atom-level couplings?
>
> We agree that extending DynaProt to atomic resolution or to protein ligand complexes is an exciting direction for future work. The main limitation at present is data availability. To our knowledge, there is no large scale MD dataset of protein-ligand complexes comparable to ATLAS in size and even ATLAS itself contains only ~1400 trajectories. At this sample size, empirical covariance of all-atom coordinates would probably be noisy, making label construction unreliable. This is the primary reason we restrict DynaProt to calpha coordinates, where the marginal 3 by 3 covariances can be modeled more robustly.
>
> That said, if high quality and large scale all-atom MD data becomes available, extending the marginal prediction to atomic resolution is conceptually straightforward, since they are predicted independently. Modeling atom-atom couplings, however, as the reviewer points out would involve much higher dimensionality for full dense covariances. Perhaps a sparse or block coupling could be imposed, but we view this as a promising direction for future work once appropriate datasets exist.
>
> We appreciate the reviewer’s thoughtful feedback and hope our detailed responses and additional experiments help raise the score.

---

> > ### Comment · Reviewer_6FkX · 2025-11-27
> >
> > I appreciate the response and further experiments from the authors. I have some questions regarding the response:
> >
> > ##  DynaProt-M and DynaProt-J
> > > We train them separately because the two tasks have different practical use cases: marginal Gaussians do not require pairwise features and keeping DynaProt-M standalone makes it even more stable and lightweight (< 1M params).
> >
> > The marginal Gaussian for each residue arises from cooperative motions among residues, so incorporating pairwise features or jointly training with DynaProt-J appears beneficial. It may seem unintuitive that you need to train two separate models instead of one.
> > Could you clarify whether you have additional experimental evidence or practical constraints that prevent doing so?
> >
> > > For example, we imagine DynaProt-M could be utilized as a PyMol plugin where the marginal gaussian blobs could be predicted and overlaid almost instantly upon loading in a structure. You are right that single multi head model is certainly feasible, but we chose the separation mainly to reduce unnecessary computation when only marginal predictions are desired.
> >
> > I don't see any technical obstacle to use only the backbone + marginal prediction head as a light inference module, even if the model is jointly trained with DynaProt-J. The question is more about the training itself and whether jointly training DynaProt-M and DynaProt-J would provide better or worse performance, as the author did not provide relevant ablation studies.
> >
> >
> > ## New cryptic pocket experiment from BioEmu protein list
> >
> > I appreciate the additional results on the expanded tests. However, the cosine similarity shows only the direction of residue motion and ignores the magnitude. These marginals also do not verify whether the cooperative movements between residues are correctly captured, which is important for apo -> holo transitions.
> >
> > The ensemble sampling demonstrated for 1ADE_A is more intuitive in this regard and could serve as a practical evaluation metric for the expanded set of cases.

---

> > > ### Author Response · Authors · 2025-12-01
> > >
> > > We thank the reviewer for their continued engagement, and although they are no longer active in the discussion phase, we provide this response to ensure the ACs have a transparent record of our responses.
> > >
> > > ### **DynaProt-M and DynaProt-J additional experiment**
> > > > ... whether jointly training DynaProt-M and DynaProt-J would provide better or worse performance.
> > >
> > > We thank the reviewer for raising this point and agree that a unified model is certainly feasible. But, our decision to keep DynaProt-M and DynaProt-J separate was driven also by empirical observations rather than just maintaining the parameter efficiency of DynaProt-M. Early in development, joint training led to instabilities because the joint covariance head is very high dimensional and enforcing positive definiteness constraints during optimization proved difficult. In practice, this separation does not impose additional burden on users as our codebase loads and runs both models seamlessly when needed.
> > >
> > > That said, to address the reviewer’s question directly, **we have now retrained a single joint model combining DynaProt-M and DynaProt-J.** Empirically, we observe clear performance degradation and substantially more unstable loss curves. The results are summarized below:
> > >
> > >
> > > | Metric            | DynaProt-M (regular) | DynaProt (jointly trained) |
> > > |-------------------|------------------------|-----------------------------|
> > > | RMWD Var (↓)      | **1.18** | 1.32 |
> > > | Sym KL Var (↓)    | **0.91** | 1.33 |
> > > | RMSF r (↑)        | **0.87** | 0.82 |
> > >
> > >
> > >
> > > ### **New cryptic pocket experiment from BioEmu protein list**
> > > > The ensemble sampling demonstrated for 1ADE_A is more intuitive ... could serve as a practical evaluation metric for the [BioEmu] cases.
> > >
> > > Our experiment in Appendix A.4 (Fig. 9) focuses on the alignment of DynaProt’s marginal Gaussian directions with BioEmu annotated cryptic pocket opening vectors because **this quantity reflects DynaProt’s main capability: fast residue-level directional anisotropy**. These marginals are intended as a lightweight dynamics readout, which is why we did not originally include ensemble sampling for this experiment.
> > >
> > > However, the reviewer brings up a good point about checking the ensembles as well, so we have now performed an evaluation analogous to our 1ADE_A study in the previous response. For each BioEmu cryptic pocket protein, we supplied the apo structure to DynaProt and generated ensembles; using BioEmu’s coverage criterion (0.01%) and a 2 A threshold, the holo structure was recovered in 10 cases.
> > >
> > > Together, both analyses indicate that DynaProt provides both **accurate directional signals and sometimes even meaningful conformational variability for apo/holo transitions**.

---

### Meta-Review · Area_Chair_LZYU · 2026-01-08

**Summary:**

The major concerns are as follows.

1. The limitation lies in the Gaussian approximation and reliance on input structures.  A Gaussian hierarchy simplifies learning and sampling, and it can not reflect the thermodynamics of MD data.
2. Case study and more completed ablation studies are needed to make the paper more comprehensive and convincing.
3. The rationality of using a network to learn the covariance matrix is questionable.
4. Lacks an explanation for the design of the covariance matrix.
5. The model was trained on the ATLAS dataset comprising 100 ns simulations is wholly inadequate. As a result, the model may have predominantly learned local, transient thermal fluctuations while exhibiting limited capacities.
6. The authors assume that the provided protein structure is in a low-energy state, but there is no discussion of how the model behaves when the input structure is not in such a state.

**Reviewer Concerns:**

There are two concerns that the reviewers are still have.

1. The rationality of using a network to learn the covariance matrix is questionable.
2. Lacks an explanation for the design of the covariance matrix.

**Reviewer Scores:**

According to the discussion between the authors and the reviewers, I think that the original scores should be 6 4 4 8. Note that, the reviewer (with score 4) claim that his concerns are not be fully addressed by the authors..

---

### Decision · Program_Chairs · 2026-01-26

Accept (Poster)